# Thromboinflammatory Processes at the Nexus of Metabolic Dysfunction and Prostate Cancer: The Emerging Role of Periprostatic Adipose Tissue

**DOI:** 10.3390/cancers14071679

**Published:** 2022-03-25

**Authors:** Ibrahim AlZaim, Aya Al-Saidi, Safaa H. Hammoud, Nadine Darwiche, Yusra Al-Dhaheri, Ali H. Eid, Ahmed F. El-Yazbi

**Affiliations:** 1Department of Biochemistry and Molecular Genetics, Faculty of Medicine, American University of Beirut, Riad El-Solh 1107 2020, Beirut 11-0236, Lebanon; ima39@mail.aub.edu (I.A.); nd03@aub.edu.lb (N.D.); 2Department of Pharmacology and Toxicology, Faculty of Medicine, American University of Beirut, Riad El-Solh 1107 2020, Beirut 11-0236, Lebanon; ama255@mail.aub.edu; 3Department of Pharmacology and Therapeutics, Faculty of Pharmacy, Beirut Arab University, Beirut 11-5020, Lebanon; shamoud@bau.edu.lb; 4Department of Biology, United Arab Emirates University, Al-Ain 15551, United Arab Emirates; yusra.aldhaheri@uaeu.ac.ae; 5Department of Basic Medical Sciences, College of Medicine, QU Health, Qatar University, Doha 2713, Qatar; 6Department of Pharmacology and Toxicology, Faculty of Pharmacy, Alexandria University, Alexandria 21521, Egypt; 7Faculty of Pharmacy, Alamein International University, Alamein City 5060310, Egypt

**Keywords:** periprostatic adipose tissue, prostate cancer, metabolic dysfunction, obesity, adipokine

## Abstract

**Simple Summary:**

As overweight and obesity increase among the population worldwide, a parallel increase in the number of individuals diagnosed with prostate cancer was observed. There appears to be a relationship between both diseases where the increase in the mass of fat tissue can lead to inflammation. Such a state of inflammation could produce many factors that increase the aggressiveness of prostate cancer, especially if this inflammation occurred in the fat stores adjacent to the prostate. Another important observation that links obesity, fat tissue inflammation, and prostate cancer is the increased production of blood clotting factors. In this article, we attempt to explain the role of these latter factors in the effect of increased body weight on the progression of prostate cancer and propose new ways of treatment that act by affecting how these clotting factors work.

**Abstract:**

The increased global prevalence of metabolic disorders including obesity, insulin resistance, metabolic syndrome and diabetes is mirrored by an increased incidence of prostate cancer (PCa). Ample evidence suggests that these metabolic disorders, being characterized by adipose tissue (AT) expansion and inflammation, not only present as risk factors for the development of PCa, but also drive its increased aggressiveness, enhanced progression, and metastasis. Despite the emerging molecular mechanisms linking AT dysfunction to the various hallmarks of PCa, thromboinflammatory processes implicated in the crosstalk between these diseases have not been thoroughly investigated. This is of particular importance as both diseases present states of hypercoagulability. Accumulating evidence implicates tissue factor, thrombin, and active factor X as well as other players of the coagulation cascade in the pathophysiological processes driving cancer development and progression. In this regard, it becomes pivotal to elucidate the thromboinflammatory processes occurring in the periprostatic adipose tissue (PPAT), a fundamental microenvironmental niche of the prostate. Here, we highlight key findings linking thromboinflammation and the pleiotropic effects of coagulation factors and their inhibitors in metabolic diseases, PCa, and their crosstalk. We also propose several novel therapeutic targets and therapeutic interventions possibly modulating the interaction between these pathological states.

## 1. Introduction

Despite a long-term decline in prostate cancer (PCa) patient mortality, PCa currently represents one of the leading cancers in men in terms of incidence, morbidity, and mortality [1]. Indeed, this is mirrored by an increased global prevalence and augmented economic burden of metabolic diseases including insulin resistance, metabolic syndrome (MetS), obesity, and diabetes, which are states of chronic low-grade inflammation [2,3]. Although seemingly distinct, accumulating evidence highlights associations between metabolic disturbances, particularly those culminating in obesity, and PCa development, progression, and worse prognosis [4]. Such associations are underpinned by several factors, the most prominent of which is adipose tissue (AT) inflammation and the disruption of metabolic homeostasis. AT expansion occurs in response to an imbalance in energy acquisition and energy expenditure favoring energy storage in adipocytes in the form of triglycerides, which is precipitated by hyperinsulinemia and drives insulin resistance and its accompanying metabolic derangements [5,6,7]. The diametric expansion of hypertrophied adipocytes beyond the diffusion potential of oxygen results in localized hypoxia, particularly in the absence of adequate compensatory angiogenesis [8,9,10]. This hypoxic state results in adipocyte death and the release of proinflammatory cytokines and chemokines, initiating AT inflammation, characterized by the infiltration of heterogenous populations of immune cells, the most prominent of which are macrophages [11]. Indeed, single cell and single nucleus transcriptomics have revealed marked heterogeneity in not only AT stromal cells, including resident and infiltrating immune cells, endothelial cells, and adipocyte progenitors, but also mature adipocytes exhibiting differential responsiveness to insulin in states of health and disease [12,13,14,15]. This extensive heterogeneity between adipose depots and the adipocytes of a given depot itself results in the emergence of a differential, depot-specific susceptibility to inflammation [11,16]. Ample evidence links increased visceral adiposity to a worse prognosis of PCa. Nevertheless, an under-studied AT that is particularly pertinent to the etiology of PCa is the periprostatic adipose tissue (PPAT) that is situated in the proximity of the prostate [17]. Indeed, PPAT dysfunction and inflammation not only amplify rudimentary neoplastic alterations of the healthy prostate, but also promote PCa progression, metastasis, and resistance to chemotherapy [17,18,19,20,21,22]. Compared to other adipose depots, the cellular and molecular heterogeneity as well as the thermogenic potential of PPAT remain largely uninvestigated.

Despite the emerging molecular mechanisms linking the complex etiology of AT dysfunction to the various hallmarks of PCa, thromboinflammatory processes implicated in this cross-talk have not been thoroughly investigated [23,24]. Indeed, PCa and the whole spectrum of metabolic diseases represent hypercoagulable states, i.e., states that further exacerbate AT inflammation and thus, prime PCa initiation and progression. Importantly, hypercoagulability is found in overweight patients who do not present diagnostic criteria of MetS and increases with the severity of obesity [25]. In addition to their pro-tumorigenic roles, factors of the coagulation cascade and their downstream signaling pathways have been shown to drive AT inflammation [24,26]. Particularly, thrombin and FXa represent valuable targets for pharmacological interventions as the pleiotropic effects of both coagulation factors mostly derive from their proteolytic cleavage of protease-activated receptors (PARs) [26,27]. As such, it becomes pivotal to elucidate the deleterious thromboinflammatory processes possibly occurring in the PPAT, a fundamental microenvironmental factor of the prostate. Here, we highlight key mechanistic links between PPAT dysfunction, PPAT secretome alterations, and PCa in light of the thromboinflammatory processes and the pleiotropic effects of coagulation factors and their pharmacological inhibitors. We also propose novel therapeutic and pharmacological interventions that warrant further investigation and are suggested to modulate the interaction between PCa and the spectrum of metabolic diseases.

## 2. Search Strategy

For this review, we searched the published literature in PubMed, EMBASE, and Google Scholar from their inception dates to 2022 for papers written in English on the pathological mechanisms linking metabolic disease-associated AT dysfunction and PCa. We mainly looked for articles discussing fundamental molecular mechanisms as well as clinical data showing correlation/association to synthesize the possible hypotheses suggesting PPAT dysfunction to be at the core of PCa development and enhanced aggressiveness. We used key words and phrases such as “Periprostatic adipose tissue and prostate cancer”, “Obesity and prostate cancer”, “Prostate cancer and coagulation”. We also looked for similar research, examined the reference lists of published narratives and systemic reviews, and searched for possible mechanistic explanations among references of similar articles.

## 3. Metabolic Syndrome, Obesity, Diabetes, and Prostate Cancer: States of Hypercoagulability

MetS comprises a cluster of disorders that adversely influence homeostatic cellular and organismal pathways. Weight gain and obesity have led to an increase in the number of subjects with MetS and have become major public health problems, now representing the second most common cause of preventable mortality globally [28,29]. Importantly, it is now known that the crux of the MetS and of obesity is AT dysfunction, which in turn predisposes patients to diabetes, its complications, as well as to cardiovascular diseases [30,31]. Apart from these endocrine and metabolic disturbances, MetS is frequently accompanied by a prothrombotic state that is reflected by AT dysfunction and alterations in the coagulation system [26,32,33]. Substantial clinical evidence emphasizes the upregulation of TF pathway in states of obesity and MetS [32,33,34]. Indeed, obese subjects have been shown to possess elevated levels of circulating vWF, TF, FVII, FVIII, and fibrinogen, while at the same time, displaying an increased secretion of PAI-1 and TAFI [24]. Moreover, increased visceral adiposity has been suggested to be an independent determinant of hypercoagulability in morbidly obese patients, as well as an instigator of shortened blood coagulation times in genetically induced and diet-induced obesity (DIO) animal models [35,36]. Likewise, central obesity is associated with high concentrations of factor VII and X, whereas no alteration in the antithrombin levels were seen in males with MetS [37]. However, inconsistent and conflicting data are reported on coagulation inhibitors. For instance, the inhibitor level of the TF pathway was found to be either increased or unaltered in MetS [38]. Nevertheless, chronic inflammation and insulin resistance are correlated with high levels of fibrinogen and fibrin formation, which reflects impaired fibrinolysis [39,40,41]. Obesity is also found to trigger increases in the plasma levels of FVII [32], thrombin-antithrombin III (TAT) complex [33], and increased activity in the circulating monocyte TF [34]. Factor VII is also correlated to body mass index and triglyceride levels [42]. Therefore, the increase in both TF and factor VII boosts the activation of the coagulation cascade. Indeed, macrophages and monocytes display relevant TF coagulation activity. Following either extracellular adenosine triphosphate (ATP), FFA-mediated stimulation of macrophage P2X7 receptor or complement fixation to monocyte surfaces, TF becomes activated, thereafter generating procoagulant microparticles (MPs), which further trigger the release of proinflammatory cytokines [43,44,45,46]. AT-infiltrating macrophages stimulate the production of free fatty acids (FFAs) that in turn induce TF transcription downstream of Toll-like receptor (TLR) signaling in addition to the stimulation of injury mediators and Jun N-terminal kinase (JNK), which provokes subsequent TF activation [47,48,49]. In fact, obesity increases the risk of a venous thromboembolism secondary to increased thrombin generation, decreased fibrinolysis and platelet hypercoagulability [50,51].

Additionally, AT is a major source of TF which is known to be augmented in states of low-grade chronic inflammation [49,52]. Moreover, diabetic patients show exaggerated signs of hypercoagulation in comparison to nondiabetic patients [53,54]. Indeed, diabetic patients have elevated levels of circulating TF, which heightens the severity of microvascular diseases in these individuals [55,56]. Weight loss in obese patients with MetS have been associated with a significant reduction in the thrombin-generation potential and circulating levels of soluble TF and factor VII [57,58]. These observations are supported by the emergence of prothrombotic states in animal models of metabolic dysfunction. Obese mice show accelerated arterial thrombosis which is associated with high levels of TAT [59]. Moreover, obese mice exhibit increased TF activity in adipocytes and adipose-infiltrating macrophages [60,61]. Consistent with clinical observations, caloric restriction and weight loss in obese mice decrease plasma FVII levels and AT inflammation [62].

Chronic pathological inflammation, a common hallmark of metabolic diseases, represents a major factor in the development of various malignancies, such as PCa [63]. Accumulating evidence indicates that adiposity is associated with a higher risk of advanced PCa [64]. Indeed, the newly recognized crosstalk between inflammatory processes, cancer and alerted hemostasis is thought to underpin cancer development and progression. Cancer creates a highly prothrombotic state that is evidenced by cancer-associated venous thromboembolic events representing a major cause of morbidity and mortality in patients with malignancy [65], to counteract which, anticoagulants are frequently incorporated into the treatment guidelines for cancer patients [66,67]. Moreover, research implicates components of the hemostatic system in the early neoplastic alterations pivotal for tumorigenesis and tumor growth [68]. Indeed, there exists a reciprocal interaction between cancer and components of the hemostatic system, by which cancer promotes thromboembolism, and the aberrant activity of components of the hemostatic system enhances cancer development, progression, and metastasis [69]. This is supported by studies providing evidence that chronic conventional anticoagulation is associated with a decreased incidence of certain cancers [70,71]. A wide body of evidence suggests that full length TF (flTF) on subendothelial cells is highly expressed in a variety of solid tumors and tumor vasculature and is a result of well-defined upstream events occurring during oncogenic transformation [72,73,74,75,76,77]. Furthermore, the soluble variant of flTF stimulates angiogenesis-independent of FVIIa [78]. Tumor cells expressing TF release TF-positive MPs, which are highly procoagulant [79,80,81]. These MPs are also found to bind to sites of vascular injury and enhance thrombosis, suggesting that MPs could serve as a biomarker to categorize patients with cancer at high risk of thrombosis [82,83]. Increased MPs TF activity is correlated with a worsened cancer stage, grade and survival rates in patients with metastatic pancreatic cancer [84]. Moreover, TF is found to activate PAR-dependent tumor cell behavior, regulate integrin function and facilitate tumor angiogenesis [72,85].

Several studies evaluated the expression of different coagulation factors in benign and malignant prostate tissues. It was shown that both the prostatic epithelium and the stroma express thrombin, the generation level and procoagulant activity of which is associated with more advanced disease and a worse prognosis of PCa [86]. Thrombin is a protease possessing a myriad of substrates including PAR-1, PAR-2, and PAR-4 in addition to its hemostatic substrate, fibrinogen [27]. The enhanced generation of thrombin in metabolic diseases is thought to drive neoplastic changes in the healthy prostate, partially through its activity on PARs. Indeed, one study showed that PAR-1, but not PAR-3 or PAR-4 expression was detected in several human PCa cell lines and immortalized human prostate epithelial cells [87], while another study demonstrated that the expression levels PAR-1, PAR-2, and PAR-4 were higher in PCa compared to normal glands [88]. It was also demonstrated that PAR-1 expression is increased in advanced-stage PCa and is localized to endothelial cells in the vascular network of prostate tumor areas [89]. The direct effects of components of the coagulation cascade, particularly thrombin, and its downstream proteolytic substrates in PCa have been reviewed elsewhere [69], and extend beyond the realm of our current focus. Here, we highlight the possible molecular mechanisms linking AT thromboinflammatory processes, with particular emphasis on the PPAT, and how these processes may ultimately drive PCa development and progression.

## 4. Thromboinflammatory Processes Drive Adipose Tissue Inflammation

A hypercoagulable state is believed to contribute to obesity-associated diseases such as diabetes, cardiovascular diseases, and cancer. Nevertheless, mechanistic links between both pathophysiological phenomena are not yet well established. The pleiotropic activities of coagulation factors that extend beyond coagulation are thought to drive metabolic diseases via their activities on metabolically active tissues such as the AT [24]. In vitro studies demonstrated that thrombin increases adipocyte secretion of the proinflammatory cytokines IL-1β, IL-6, MCP-1, and TNF-α, along with the growth factor VEGF [90]. Importantly, thrombin-induced inflammation has long been shown to upregulate the expression of angiogenic factors, matrix-degrading proteases, and adhesion molecules, in addition to cytokines that promote tumor cell proliferation, invasion, angiogenesis, and metastasis [91]. Indeed, inhibiting thrombin activity with dabigatran decreases proinflammatory cytokine production in HFD-fed female LDL receptor-deficient mice and decreases M1 macrophage polarization without significantly reducing adipocyte hypertrophy [92]. This could partly be explained by the inhibition of thrombin-mediated induction of M1 macrophage polarization, a process that is mediated in part by the cleavage of PAR-1 and possibly PAR-4 and downstream PI3K/AKT and NF-κB signaling [93,94]. Moreover, a 4-week administration of argatroban to db/db obese diabetic mice was shown to reduce fasting plasma glucose and enhance insulin sensitivity [95]. Argatroban treatment also reduced adipocyte size, macrophage infiltration and the expression of MCP-1, IL-6, and factor VII in the AT [95]. Therefore, the enhanced adiposity-induced, factor VII-mediated generation of thrombin is thought to drive AT insulin resistance and macrophage infiltration. Indeed, it was shown that FVII secretion from adipocytes occurs following TNF-α stimulation, likely via pathways involving mitogen-activated kinases and NF-κB activation [96]. Additionally, β-adrenoreceptor agonism stimulates FVII secretion from adipocytes via the cyclic AMP/PKA pathway [96].

Yet, other studies found that the enhanced thrombin generation was modulated by diet, independent of the metabolic status of rats and humans including insulin resistance, obesity and serum coagulation factor levels [97]. Nevertheless, this does not exclude the possibility that HFD-induced increases in thrombin generation can drive DIO and the complications of AT inflammation. Indeed, extravascular fibrin deposits were recently identified in the white adipose tissue (WAT) of HFD-fed mice and obese individuals [98]. Mice carrying a mutant form of fibrinogen that is incapable of binding to the leukocyte αMβ2-integren were not only resistant to the diet-induced gain of adiposity, but also had significantly diminished AT inflammation and macrophage infiltration [98]. Contrastingly, homozygous thrombomodulin mutant mice, characterized by an elevated procoagulant activity of thrombin, exhibited a more pronounced diet-induced weight gain [98]. Moreover, the pharmacological inhibition of thrombin activity with dabigatran limited the development of DIO in mice and halted its progression [98,99]. This collectively provides evidence that thrombin activity is central to HFD-induced adiposity, weight gain, and AT inflammation, and that its pharmacological inhibition with direct thrombin inhibitors offers a therapeutic approach to curb the complications of such metabolic derangements. Indeed, tissue factor mediates the FVIIa-dependent activation of PAR-2, which was shown to drive DIO and its complications as nonhematopoietic cell TF-FVIIa-PAR-2 signaling promotes obesity [61]. Mechanistically, adipocyte tissue factor cytoplasmic domain-dependent FVIIa signaling suppresses Akt phosphorylation, leading to the dysregulation of key metabolic regulators, which were reversed by the pharmacological blockade of adipocyte TF. Along the same lines, mice either lacking PAR-2 or the cytoplasmic domain of TF were protected against HFD-induced weight gain and insulin resistance [61]. Moreover, the genetic ablation of TF-PAR2 signaling in hematopoietic cells or the pharmacological inhibition of macrophage TF significantly reduced AT macrophage infiltration and inflammation [61]. Intersections between procoagulant and inflammatory pathways in AT under conditions of obesity and metabolic syndrome are represented in Figure 1.

## 5. The Molecular Specificity of Periprostatic Adipose Tissue

Research related to the high resolution cellular characterization of periprostatic adipocytes is lacking. Accumulating data emphasize the morphological, cellular, and molecular heterogeneity of distinct adipose depots [11]. WAT, which specializes in lipid storage and adipokines production, is composed of unilocular mitochondria-poor adipocytes, in addition to vascular stromal cells, which include adipose progenitors, endothelial, and immune cells [11]. WAT is subcategorized into visceral (VAT) and subcutaneous (SAT) adipose tissue. Alternatively, thermogenically active brown adipose tissue (BAT) comprises the stromal vascular fraction, multi-locular, mitochondria-rich adipocytes that specialize in energy dissipation [100]. PPAT has been considered as a VAT despite its extraperitoneal localization. Nevertheless, an extensive comparison between PPAT, VAT, and SAT depots has not yet been conducted. Therefore, coculture systems utilizing primary adipocytes or preadipocytes and PCa cells differentiated in vitro must be interpreted cautiously since they do not necessarily recapitulate the specific characteristics of adipocytes belonging to the PPAT. In comparison to subcutaneous adipocytes, periprostatic adipocytes are smaller and are more sensitive to isoproterenol-stimulated lipolysis [101]. Additionally, PPAT was shown to have a higher content of adipose-derived stem cells in comparison to VAT [102]. Nevertheless, in vitro differentiated, as well as isolated primary adipocytes remain viable tools in the study of complex tumor-AT crosstalk, as they faithfully mimic in vivo pathological alterations [103,104].

Importantly, the thermogenic potential of the PPAT has not been investigated. As these adipocytes morphologically resemble white adipocytes, their expression of thermogenic markers, including uncoupling protein 1 (UCP1) has not been assessed. Additionally, the hallmarks of alternative thermogenic pathways such as creatine futile cycling have only recently been identified. Downstream of β3-adrenergic signaling, UCP1 uncouples mitochondrial oxidative phosphorylation from the production of ATP while dissipating the protonmotive force, thus releasing energy in the form of heat. Alternatively, less-efficient thermogenic pathways downstream of β3-adrenergic signaling include UCP1-independent proton leak through the mitochondrial ADP/ATP carrier, which is initiated at a high membrane potential, the lipolysis/re-esterification cycle that is based on the energy demand of triacylglycerol synthesis, the SR/ER calcium ATPase pump and phospholamban-dependent calcium cycling, and futile creatine cycling [11,100]. Moreover, the browning potential of PPAT has not been assessed in response to β3-adrenergic stimulation. Indeed, mitochondrial patch clamp assays of beige adipocytes revealed UCP1-positive and UCP1-negative cells, both of which exhibit creatine futile cycling [105]. Therefore, a tentative investigation of PPAT thermogenic potential is warranted, particularly due to the presence of sympathetic nerves embedded in the PPAT [106,107,108]. Investigating PPAT thermogenic capacity in health and disease is an emerging field of research owing to the anticipated roles for PPAT browning on the malignant behavior of PCa [109]. This is of particular interest as the UCP1-dependent and UCP1-independent uncoupling of mitochondrial respiration and oxidative phosphorylation are believed to drive the aberrant augmentation of the mitochondrial oxygen consumption rate and oxygen demand, possibly predisposing PPAT to hypoxia and if already present, exacerbating the local hypoxic state, driving further metabolic dysfunction [110,111].

## 6. Periprostatic Adipose Tissue: A Microenvironmental Contributor to Prostate Cancer

The thickness of PPAT has been associated with enhanced PCa aggressiveness and the dysregulated secretome, which is thought to instigate PCa progression in a paracrine manner. Emerging evidence implicates cellular components of WAT in the progression of PCa [17]. PPAT covers the subperitoneal region and is bordered by the anteriorpubic symphysis, lateral obturator muscles, and the posterior rectoprostatic fascia [112]. Although the PPAT shares common vasculature with the prostate, they are separated by a fibromuscular capsule. Nevertheless, one-third of the prostate anterior is in direct contact with the surrounding AT [112,113]. This proximal localization allows for a profound crosstalk by which PPAT dysfunction, as a result of metabolic impairment, possibly contributes to neoplastic alterations of the healthy prostate and enhances the aggressiveness of PCa.

PPAT thickness, or volume, whether normalized to that of the prostate or not, have been demonstrated as independent predictors of PCa development, grade of PCa at biopsy, and tumor response to androgen deprivation therapy [114,115,116]. Indeed, there exists a significant correlation between increased PPAT volume and shortened progression-free survival in men with PCa [117]. Particularly, several studies demonstrated a significant positive correlation between PPAT size and, not only PCa aggressiveness, but also the development of castration-resistant PCa irrespective of treatment naivety and lymph node metastasis [118,119,120,121]. Nevertheless, these studies relied on the body mass index (BMI) as a marker for general obesity, and on PPAT size as a measure of its contribution to the hallmarks of PCa, rather than examining the inflammatory status of visceral fat, and particularly that of the PPAT. In fact, one study in PCa patients receiving brachytherapy showed no significant correlation between PPAT size and fat density, and PCa aggressiveness where patients were subcategorized into normal weight, overweight, and obese subjects [122]. BMI was also not correlated with PPAT volume and density in a retrospective study of PCa patients scheduled to receive radiation therapy [123]. Moreover, one study showed that a higher PPAT ratio was significantly correlated with a higher Gleason score in PCa patients irrespective of BMI, serum prostate-specific antigen, and age [124]. However, other studies have shown that, in addition to PPAT thickness, BMI and serum prostate-specific antigen levels significantly correlate with the Gleason score in PCa patients [125]. Indeed, correlations between PPAT and BMI have yielded inconsistent results owing to the fact that BMI does not represent a qualitative indicator of body composition or metabolic health and does not discriminate between visceral and subcutaneous adiposity [19]. It therefore becomes evident that the inflammatory status of the PPAT rather than its dimensional measurements should be accounted for when assessing its correlation with PCa-associated parameters.

As previously described, AT expansion ensues in response to a prolonged positive energy balance in order to accommodate the increased requirement of fat storage. Hypertrophic expansion of adipocytes results in the formation of local hypoxia, which leads to cellular damage, cellular death, and eventually to inflammation [5,6,7]. As epidemiological and clinical studies implicate increased adiposity as a driver of PCa development, progression, and metastasis, investigation into inflammatory processes underpinning PPAT inflammation becomes essential, particularly with the accumulating evidence of HFD-induced prostatic intraepithelial neoplasia and PCa mainly through the activation of inflammatory processes [126,127,128]. Indeed, it was shown that PPAT inflammation in PCa patients, evidenced by increased crown-like structure formation is associated with higher BMI, larger adipocytes, and a higher Gleason grade [129]. PPAT inflammation was also found to correlate with higher levels of circulating insulin, triglycerides, and leptin, which are prominent hallmarks of the MetS. Murine DIO models exhibit pronounced PPAT inflammation secondary to AT expansion, evidenced by the augmented expression of CD68, MCP1, and TNF-α, and the increased formation of CLS, which is consistent with enrichment in inflammatory response pathways [103,130]. Despite ample evidence of obesity-associated PPAT inflammation-induced promotion of PCa, cancerous hallmarks of mouse xenografts of patient-derived, moderate-grade localized PCa were not altered by obesity in HFD-fed mice [131]. Although coculture systems of stromal and epithelial cells demonstrated a pro-tumorigenic effect of human PPAT-secreted factors, co-grafting human PPAT with PCa in patient-derived xenograft mice revealed no tumorigenic acceleration in comparison to the PCa graft alone [131]. These findings prompt the requirement for further investigation into the mechanisms by which increased adiposity and obesity promote PCa in light of PPAT dysfunction. Figure 2 summarizes some of the suggested pathways mediating this effect.

## 7. Periprostatic Adipose Tissue as a Lipid Provider to Prostate Cancer

Adipocytes possess the capacity to release FFAs from stored lipids in response to increasing environmental energy demands, either through lipolysis or via extracellular vesicular transport, particularly through exosomes [132,133]. In the context of the thrombogenic state associated with AT inflammation and PCa, thrombin was shown to enhance adipocyte lipolysis in vitro through the activation of lipoprotein lipase and is thought to regulate adipocytic lipid metabolism [134,135]. In addition to being substrates for enhanced PCa lipogenesis, these FFAs are taken up by proximal cancer cells and are utilized as a pivotal energy source driving cancer progression and metastasis, blocking either the fatty acid transporter CD36, or the membrane-bound transcription factor, sterol-regulatory element binding protein (SREBP) and halts disease progression and metastasis in different preclinical models of PCa [136,137,138]. Particularly, PPAT represents a preeminent source of FFAs for PCa cells which drive PCa development and aggressiveness, a process amplified by obesity [139]. Moreover, it was shown that increased saturated fatty acids in the serum of obese Zucker rats reinforce the procoagulant state in MetS [140]. Indeed, using magnetic resonance spectroscopically the FFA composition of the PPAT was analyzed, revealing a differential association with hallmarks of PCa aggressiveness in PCa patients [141]. Extracapsular extension is associated with an elevated level of monounsaturated and a reduced level of saturated fatty acids in the PPAT, while it appears that the unsaturated to saturated fatty acids ratio exhibit a modest, inverse correlation with the Gleason score [141,142]. These alterations in PPAT fatty acid composition might well explain the enhanced proliferative capacity of PCa and endothelial cells treated with explant tissue culture secretions of PPAT obtained from obese patients undergoing surgical prostate removal [142]. It was also shown that low levels of both the essential polyunsaturated fatty acids, linoleic and eicosapentaenoic acids in PPAT were associated with PCa aggressiveness [143]. Moreover, the migration potential of PPAT-derived fatty acid extract-supplemented PCa cells were found to be inversely correlated with PPAT linoleic acid content [143]. These polyunsaturated fatty acids were shown to impart an anticancer effect by modulating the bidirectional interaction between Ca^2+^-activated K^+^ channel SK3, which amplifies calcium entry and cellular migration, and the epithelial-mesenchymal transition transcription factor Zeb1 [144].

Indeed, a bidirectional crosstalk has been demonstrated in PCa cells invading PPAT, which in turn promotes tumor cell invasion [104]. In an in vitro coculture system of PCa cells and adipocytes, tumor cells were shown to induce adipocyte lipolysis and FFA release. Released FFAs are taken up and stored by PCa cells, which acquire enhanced invasive characteristics. Importantly, this induces the expression of NOX5, an isoform of NADPH oxidase, and the subsequent production of intracellular ROS [104]. Increased Intracellular ROS activates the HIF-1α/MMP-14 signaling pathway, which accounts for enhanced invasiveness [104]. Of clinical relevance, tumor-surrounding adipocytes in obese patients are more prone to activating the HIF-1α/MMP-14 signaling pathway, and the expression of NOX5 and MMP-14 is upregulated in the invasive front of tumors, the side proximal to the PPAT [104]. Similarly, an increased PPAT area in DIO mice was associated with increased NOX2 and TNF-α expression, along with decreased SOD1 expression [104]. Additionally, it was recently demonstrated that lipophagy plays a crucial role in PCa progression [145]. Microenvironmental FFAs taken up by PCa cells are stored in lipid droplets, the lipolysis of which fuels PCa progression. Indeed, markers of lipid droplets and autophagy in prostate tissues are associated with disease aggressiveness in PCa patients, and are increased in extraprostatic areas that are in direct proximity to the PPAT [145]. Additionally, it was shown that high fat diets, particularly saturated fat-rich diets not only promote the synergistic pathological transformation mediated by Src kinase and androgen receptors, but also enhance the proliferation of Src-mediated xenografts [146]. Furthermore, in vitro palmitate treatment upregulated the biosynthesis of palmitoyl-CoA, enhanced Src-dependent mitochondrial oxidation, and increased Src-mediated downstream signaling, including the activation of MAPK and focal adhesion kinase [146]. Nevertheless, although adipocytes belonging to the PPAT of PCa patients undergoing total prostatectomy presented comparable basal lipolysis and were more sensitive to isoproterenol-stimulated lipolysis in comparison to subcutaneous adipocytes, lipid biological features of the PPAT were associated neither with PCa aggressiveness nor obesity, a finding that does not support altered PPAT lipolysis as a driver of PCa aggressiveness [101]. Further investigation into the mechanisms by which the PPAT provides PCa cells with fatty acids is required, particularly as the impact of PPAT dysfunction on PCa metabolism might offer new therapeutic opportunities.

## 8. Periprostatic Adipose Tissue Secretome and Prostate Cancer Progression: Current Evidence and Emerging Research Avenues

Pertinent to the endocrine nature of the AT, PPAT adipocytes and stromovascular cells secrete a myriad of adipokines capable of modulating PCa behavior, probably under the instruction of PCa cells [147,148,149]. Indeed, profound alterations in the PPAT epigenome, transcriptome, and secretome have been shown in lean and obese patients with benign prostatic hypertrophy (BPH), non-aggressive, and aggressive PCa [150,151,152,153]. Importantly, proliferative and metabolic pathways were differentially regulated according to the stage of disease [150,151]. Nevertheless, the low number of proteins identified in the BPH group does not allow for the derivation of exhaustive conclusions. Indeed, a higher PPAT expression of VEGF and TNF-α correlates with high grade PCa and higher Gleason scores [154]. The aggressiveness of PCa is also correlated to the increased PPAT secretion of IL-6, but not its secretion of leptin or adiponectin [155]. Importantly, increased IL-6 and decreased adiponectin expression levels in PPAT were correlated with PCa tissue [155]. Here, we highlight key findings relating to the effect of PPAT-secreted adipokines on the prostate in states of health and disease.

### 8.1. Leptin

First described as a satiety hormone, leptin is now well-recognized as a proinflammatory adipokine that binds long form of leptin receptor numerously expressed in peripheral tissues, the activation of which regulates gastrointestinal nutrient absorption, thermogenesis, lipid and glucose homeostasis, behavior, and immunity among other functions [156,157,158,159]. Moreover, leptin levels directly correlate with AT mass, where hyperleptinemia was shown to be associated with leptin resistance commonly found in obese individuals without the anorexic response [160]. Indeed, AT dysfunction, observed in conditions of chronic energy excess, MetS and obesity, further increases leptin production which in turn, due to its proinflammatory nature, augments cytokine production and T-cell proliferation, thereby amplifying AT inflammation [159,161,162]. Within the context of PCa, leptin was not consistently associated with overall PCa and there was weak evidence of a positive association between leptin and PCa aggressiveness [163]. Nevertheless, hyperleptinemia remains a preeminent feature of PCa and PCa tissues express higher levels of leptin and the leptin receptor in comparison to BPH and healthy prostates [164]. Particularly, leptin signaling enhances several cancer-promoting pathways, including cancer growth, proliferation, migration, angiogenesis, metabolism, and inflammation. As such, it was shown that blocking leptin signaling delayed the emergence of castration-resistant PCa in a murine model of PCa, as well as reduced tumor vascularity, and altered cancer cell apoptosis and energetics [165]. Therefore, blocking leptin signaling provides a potential adjuvant therapy to androgen deprivation in advanced PCa.

At the molecular level, leptin enhances PCa cell proliferation and invasiveness in a dose-dependent manner, which is associated with an increased expression of ERK1/2, PI3K, Bcl-2 and a decreased expression of Bax and c-Caspase 3 [166,167,168]. In addition to its mitogenic and anti-apoptotic effects via the downstream phosphorylation of MAPKs and the promotion of cyclin D1 [169], leptin was shown to promote angiogenesis both in vitro and in vivo by the activation of ERK1/2 [170,171]. Indeed, the incubation of androgen-independent PCa cells with leptin induces the expression of VEGF, TGF-β1, and FGF as well as cellular migration through the activation of MAPK and PI3K signaling [172]. Nevertheless, leptin was also shown to enhance PCa cell apoptosis evidenced by an increased level of caspase 3, PARP cleavage, and DNA condensation which is mediated by the balanced activation of JAK2/STAT3, p38 MAPK, and PKC signaling pathways [173]. Additionally, high leptin concentrations inhibited cellular proliferation and enhanced apoptosis in androgen-dependent PCa cell lines evidenced by increased PARP-1 cleavage and the decreased expression of p38 MAPK, p42/44 MAPK and BCl-2 phosphorylation [174]. Moreover, high leptin serum concentrations were shown to inhibit tumor growth and angiogenesis in vivo [175]. It was also reported that leptin enhances androgen independent PCa cell proliferation via the activation of MAPK, JNK and the subsequent signaling through STAT3 and Akt [176,177]. Additionally, leptin promotes PCa cell migration and epithelial to mesenchymal transition (EMT) through the stimulation of the STAT3 pathway [178]. Long-term exposure of androgen-dependent and androgen-insensitive PCa cell lines to leptin enhanced cellular proliferation, migration, and invasion through the PI3K/Akt signaling-mediated inhibition of FOXO1 nuclear translocation, which was also associated with an increased expression of cyclin D1 and a decreased expression of p21 [179]. Another mechanism through which leptin enhances PCa cell migration is through binding to the leptin receptor and activating downstream IRS-1/PI3K/Akt/NF-κB signaling, eventually leading to an increased expression of αvβ3 integrins [180].

These closely related processes of inflammation and angiogenesis are corner stones in the development and metastasis of cancers, including PCa [129,181]. Leptin was found to influence cellular differentiation and the progression of PCa among other cancers [182]. Leptin was demonstrated to enhance the cellular growth of androgen-independent, but not androgen-responsive PCa cell lines [166]. Several studies demonstrated a positive association between PCa aggressiveness and high levels of leptin in obese individuals exhibiting larger and thicker PPAT [129,151,155,183,184,185,186]. The proinflammatory state created by leptin and PPAT inflammation further creates suitable conditions for a more aggressive tumor characterized by a larger volume, higher grade, increased proliferative and invasive capacities [151,183,185]. In addition, PPAT adipocytes have been shown to be recruited by cancer-derived factors to synthesize matrix metalloproteinases, the expression of which in prostate tumors was shown to further contribute towards an aggressive phenotype [183].

### 8.2. Adiponectin

Adiponectin is the most abundant adipokine in human plasma, accounting for up to 0.05% of total plasma protein [157]. Adiponectin expression levels are inversely associated with obesity, insulin resistance, type 2 diabetes (T2D) and various malignancies [157,187]. Originally known for its insulin sensitizing properties [188,189], adiponectin has been extensively studied for its profound anti-inflammatory and pro-adipogenic properties [190,191]. Circulating adiponectin forms low, intermediate, and high molecular weight complexes that inhibit the activation of NF-κB and the subsequent production of proinflammatory cytokines [192,193]. At the cellular level, adiponectin exerts its effects by binding to its tissue-specific receptors, AdipoR1 and AdipoR2, which results in the downstream activation of AMPK, the phosphorylation of Akt and eNOS, and production of NO [194,195,196].

Although the available literature concerning the associations between serum adiponectin levels and the various facets of carcinogenesis is quite conflicting, the most accredited hypothesis is that adiponectin exerts a protective role against the development and progression of PCa [163,197,198,199]. A meta-analysis concluded that the concentration of adiponectin in PCa patients was significantly lower than in control subjects [200]. A 25-year prospective cohort clearly demonstrated that adiponectin concentrations were inversely associated with the risk of developing high grade or lethal PCa [201]. Importantly, androgen-responsive and androgen-independent PCa cell lines in addition to human healthy and cancerous prostatic tissues were shown to express AdipoR1 and AdipoR2, whose expression is tightly regulated by the metabolic milieu [202,203]. Indeed, enhanced AdipoR2 expression is associated with increased proliferation, FAS expression, and angiogenesis in human PCa [204]. One study suggested that PCa is associated with a lower expression of AdipoR1 and AdipoR2 in comparison to BPH [205]. Nevertheless, activating adiponectin receptors inhibits PCa growth in vivo through the induction of apoptosis [206]. Adiponectin exerts a plethora of anti-tumor effects, and the silencing of endogenous adiponectin promotes the proliferation and invasion of PCa cells via an enhancement of EMT [207]. Adiponectin was found to be protective against BPH, where it inhibits cell cycle G1/S-phase progression and promotes apoptosis by inhibiting the MEK-ERK-p90RSK axis of prostatic epithelial and stromal cells [208]. It was also shown that adiponectin inhibits PCa cell proliferation, an effect that is counteracted by leptin [209]. At the molecular level, its anti-proliferative and pro-apoptotic actions are mediated via the activation of several intracellular pathways including AMPK, MAPK, and PI3K/Akt [210]. However, as adiponectin activates AMPK in PTEN-deficient LNCaP PCa cells and increases mTOR activity through PI3K and Akt activation [211], it is plausible that high endogenous adiponectin may also directly stimulate the signaling pathways that enhance tumor growth. Indeed, adiponectin was shown to enhance PCa cell migration via the upregulation of α5β1 integrins and the activation of AdipoR1, p38, AMPK, and NF-κB pathways [212]. However, another study showed that adiponectin anti-proliferative activity is achieved through the activation of AMPK in PC-3 cells, which is associated with a reduction in mTOR activation [213]. The high molecular weight complex of the full length adiponectin inhibits PCa cell proliferation at sub-physiological concentrations [214]. Moreover, full length adiponectin suppresses leptin and IGF-1-stimulated androgen-dependent and androgen-independent PCa cell growth and enhances the doxorubicin inhibition of PCa cell growth [214]. Additionally, adiponectin is thought to halt cancer neovascularization through the activation of AMPK/TSC2 that inhibits mTOR-mediated activation of VEGF-A [215]. Adiponectin also inhibits ROS formation and enhances cellular anti-oxidant pathways in human PCa cells in a dose-dependent manner [216]. It is noteworthy to mention the interplay between leptin and adiponectin, as it is well documented that a metabolic dysregulation resulting in hypoadiponectinemia and hyperleptinemia favors cancer cell growth via both systemic and local mechanisms [217]. This was supported by various studies including the findings of Zhang et al. at the level of the PCa microenvironment, where leptin was positively associated with a more aggressive phenotype of PCa, while adiponectin was found to be protective against PCa aggressiveness [155]. A cross-sectional study conducted by Gucalp et al. confirmed the correlation of PPAT inflammation with high-grade PCa associated with an elevated leptin/adiponectin ratio even after BMI adjustment [129].

### 8.3. Visfatin

In the current literature, the three terms Visfatin, Pre-B cell enhancing factor (PBEF) and extracellular nicotinamide phosphoribosyltransferase (eNampt) describe the same molecule having insulin-mimetic properties [218]. Originally thought to be produced only in VAT [219], visfatin has since been shown to not only be synthesized by different adipose depots but also to be ubiquitously expressed and associated with a variety of functions in different immune cell types such as activated macrophages, lymphocytes, monocytes, or neutrophils [220]. However, there currently exists no evidence that proves PPAT to be a source of visfatin. Visfatin was shown to be an important upstream regulator of innate immunity through binding to TLR-4 and activating NF-κB signaling pathways, suggesting a crucial role for eNAMPT in inflammation [221]. Indeed, increased serum visfatin levels correlate with markers of systemic inflammation and have been reported in metabolic diseases, such as obesity and T2D [222]. Interestingly, the infiltration of inflammatory cells in AT, such as activated macrophages, a major source of visfatin, prominently increases in relation with obesity [220]. Visfatin is upregulated in several malignancies including PCa. Moreover, this overexpression has been shown to promote acquired resistance to chemotherapeutic agents such as doxorubicin, etoposide, fluorouracil, paclitaxel, and phenylethyl isothiocyanate [223,224,225]. Sun et al. demonstrated visfatin as an upstream mediator promoting PCa cell transition to an invasive phenotype [226]. Moreover, several studies showed that visfatin inhibition promotes growth inhibition, apoptosis and autophagy in human PCa cells [227,228,229,230]. Interestingly, an in vitro PCa model mimicking the levels of visfatin in obese patients exhibited a more aggressive phenotype with higher concentrations of visfatin [222]. Nevertheless, the PPAT, as a supplier of visfatin to the neighboring prostatic tissue, especially in obese phenotype, is yet to be investigated.

### 8.4. Omentin

Additionally, known as intelectin-1, intestinal lactoferrin receptor, or galactofuranose-binding lectin, omentin, is a recently identified adipokine whose levels were reported to increase in various malignancies including PCa [231,232,233,234,235]. The circulating levels of omentin have been negatively correlated with obesity, insulin resistance, and diabetes, while being positively correlated with high density lipoprotein (HDL), adiponectin, and endothelial function in serum [236,237,238,239,240]. However, an debate exists on whether omentin levels in cancer patients demonstrate a positive correlation [233,241] or negative correlation [232,242,243] with body mass index (BMI) and waist–hip ratio (WHR). Importantly, a case–control study conducted on 40 patients with PCa and 40 patients with BPH, reported significantly elevated concentrations of omentin in patients with cancer, leading the authors to propose the use of this adipokine as a noninvasive biomarker of PCa [244]. Another study confirmed the latter results with the addition that omentin was not associated with an increased likelihood of MetS in men with PCa [243].

### 8.5. Resistin

Resistin is a proinflammatory adipokine that links diabetes and obesity by promoting insulin resistance [245]. In humans, studies have revealed that resistin is expressed in AT, with monocytes and macrophages being the main sources whereas in rodents the main source are adipocytes [245,246]. The proinflammatory actions of resistin are mediated by CAP-1, a resistin receptor, with subsequent downstream activation of NF-κB in human monocytes [247]. Moreover, resistin binds TLR4 and regulates macrophage production of TNF-α and IL-6 through the activation of NF-κB [248]. Serum resistin levels are elevated in human obesity [249,250], insulin resistance [250], and T2D [251]. Steppan et al. showed that insulin sensitivity was decreased upon the external administration of resistin to mice with impaired glucose tolerance without decreasing insulin levels [252].

Human PCa cell lines have been shown to express resistin and high-grade PCa tissues exhibit a higher expression of resistin in comparison to BPH [246]. Recently, a growing body of evidence has highlighted a role of resistin in the pathogenesis of cancer and its therapeutic outcomes. Resistin has been described to increase aggressiveness and promote cancer progression directly and indirectly, by inducing invasion, metastasis, EMT, stemness, angiogenesis, and chemoresistance, in addition to regulating cell cycle progression and apoptosis [253]. Incidentally, resistin was shown by Kim et al. to directly promote PCa cell proliferation via the PI3K/AKT pathway that mediates cell survival and growth signals by phosphorylating and inactivating pro-apoptotic proteins [245,246]. Fittingly, PI3K/AKT signaling is critical to PCa cell survival and proliferation and is associated with a more aggressive PCa phenotype [254]. Taking into account that chronic inflammation is a known cause of PCa [246], the chronic low-grade sub-clinical inflammation seen in obesity which often involves macrophage infiltration in the AT is accompanied by elevated levels of resistin [245]. As previously established, PPAT was shown to be a source of various adipokines, including resistin which, in a paracrine manner, migrates into the adjacent prostatic tumor and acts as a pro-inflammatory component in the tumor micro-environment, further leading to a poor prognosis [255].

### 8.6. LCN2

As is known, Lipocalin-2, also known as neutrophil gelatinase-associated lipocalin (NGAL) is a multifunctional secreted glycoprotein, for which circulating serum levels and expression levels in the peripheral mononuclear blood cells increase in states of metabolic dysfunction [256]. Additionally, LCN2 expression in the AT is also significantly increased in animal models of obesity and diabetes and in insulin resistant and diabetic subjects [257,258]. Nevertheless, the role of LCN2 in states of metabolic dysfunction is controversial. Adipocytes abundantly express LCN2, the expression of which is promoted during adipogenesis in a C/EBP-dependent manner and is thought to drive obesity-associated insulin resistance [259]. Moreover, the reduction of insulin resistance through the in vitro treatment or in vivo supplementation of thiazolidinediones reduces adipocyte and AT LCN2 expression [257,259,260]. As is known, LCN2-treated adipocytes exhibit increased levels of PPARγ and adiponectin, while reducing LCN2 levels reflects the opposite results [260]. Indeed, LCN2 deficiency reduces PPARγ expression in adipocytes and PPARγ agonism in HFD-fed LCN2-deficient mice significantly reduces white AT de novo lipogenesis and enhances brown AT thermogenesis [261]. Nevertheless, LCN2 increases the expression of IL-6 and decreases the expression of adiponectin and PPARγ in human SAT [262]. Moreover, LCN2-treated adipocytes exhibited reduced TNF-α-induced effects on glucose uptake, PPARγ, IRS-1, and GLUT4 expression, in addition to the secretion of leptin and adiponectin [260]. Furthermore, LCN2-treated adipocytes upregulate the expression of thermogenic proteins such as UCP1, PRDM16, ZIC-1, and TBX1 and exhibit enhanced mitochondrial activity [263]. Moreover, LCN2-treated adipocytes exhibit enhanced fatty acid oxidation and energy expenditure and LCN2 postprandial circulating levels correlate with energy expenditure in HFD-fed women [264].

Obese individuals express enhanced levels of LCN2 in VAT and increased circulating levels and activity of LCN2/MMP9 complexes [265]. Importantly, LCN2 expression in VAT positively associates with inflammatory markers [265,266]. Supporting the proinflammatory role of LCN2, LCN2-deficient genetic or dietary-induced obese mice display improved insulin signaling and reduced AT inflammation irrespective of adipocyte hypertrophy, in part due to reduced 12-lipoxygenase-mediated TNF-α production [267]. Moreover, IL-1β was shown to induce LCN2 expression in cultured murine adipocytes through NF-κB and JNK2 signaling [268,269]. Indeed, INFγ and TNF-α induce adipocytic expression and secretion of LCN2 in a STAT1 and NF-κB-dependent pathways, respectively, downstream of ERK activation [270,271].

Nevertheless, another study has shown that LCN2-deficient mice exhibit potentiated DIO and insulin resistance, impaired adaptive thermogenesis, and enhanced AT inflammation [272]. The LCN2-silenced mice exhibit worsened metabolic dysfunction in diet and genetically induced models of obesity [273], while increasing LCN2 circulating levels was suggested to promote adaptive pancreatic β-cell proliferation. Indeed, LCN2 was shown to regulate BAT activation through the adrenergic-independent p38 MAPK/PGC-1α/UCP1 pathway in LCN2-defient mice, maintaining normal sympathetic innervation and activation in BAT [274]. Nevertheless, the role of LCN2 in adaptive thermogenesis remains controversial, as circulating LCN2 was suggested to possess anti-thermogenic effects through the inhibition of BAT activity in an iron-dependent manner [275]. As a target gene of retinoic acid, LCN2 was shown to be required for full ATRA-mediated induction of BAT thermogenesis, where LCN2-deficient mice exhibit impaired retinol binding protein 4 AT secretion [276]. Moreover, LCN2 regulates the non-genomic actions of retinoic acid on the activation of beige adipocytes by [277]. Moreover, aged mice that overexpress LCN2, specifically in adipocytes, exhibit smaller inguinal adipocytes, increased markers of adipogenesis, and attenuation of age-induced reduction in phosphorylated AMPK [278]. Despite these observations in LCN2-deficient mice, another study highlighted that the global ablation of LCN2 had a minor effect on obesity-associated glucose intolerance but no effect on obesity-mediated insulin resistance [279]. Other studies demonstrated no metabolic differences between normal chow and HFD-fed aged LCN2 knockout mice and postulated that the deletion of LCN2 promotes metabolic dysfunction possibly through gut microbiota dysbiosis [280,281].

High prostatic tissue LCN2 expression is associated with significantly reduced tumor differentiation and higher Gleason scores in PCa patients [282,283,284]. Moreover, higher LCN2 levels in serum and prostatic tissue of PCa patients positively correlates with invasive phenotypes [285]. Nevertheless, the predictive and prognostic utility of evaluating LCN2 expression requires further investigation. The expression of LCN2 is higher in androgen-insensitive PCa cell lines with an enhanced invasive capacity and its expression is tightly regulated by NF-κB, p53, and the AR [283,286]. Knocking down LCN2 in PC3 and DU145 cells reduces cellular proliferation, migration, invasion, and colony formation capacity, and causes cell cycle arrest, while overexpressing LCN2 in 22Rv1 cells results in the opposite observations [283]. Similar results were obtained when either knocking down or overexpressing LCN2 in C4-2B and 22Rv1 cells [284]. It was observed that LCN2 overexpression in 22Rv1 cells promoted EMT and enhanced cellular motility and invasiveness through the ERK signaling-mediated induction of SLUG expression, while knocking LCN2 down in PC3 cells yielded the opposite result [285]. Overexpressing LCN2 in vivo also promoted tumor growth via enhanced AR transcriptional activity in C4-2B and 22Rv1 cells [284]. Pathways regulating LCN2 expression in PCa remain to be investigated. Nevertheless, it was recently shown that LCN2 expression and secretion are significantly induced by TNF-α in PC-3 cells through the activation of NF-κB and JNK signaling [287]. Moreover, ER stress was shown to induce LCN2 expression in murine and human PCa cells via the activation of the unfolded protein response in an NF-κB-dependent manner [288]. Additionally, LCN2-deficient PC3 cells exhibit reduced proliferation and increased sensitivity to cisplatin-induced apoptosis [289].

### 8.7. RBP4

Retinol binding protein 4 (RBP4) and its membrane receptor, STRA6, facilitate circulating retinol transfer and coordinate cellular retinol homeostasis. In addition to its hepatic secretion, RBP4 is secreted by SAT and VAT, with a more pronounced production in the latter in states of obesity and diabetes [290]. This was associated with an increased circulating level of transthyretin, a stabilizer of circulating RBP4 that reduces its renal clearance, in obese subjects and leptin-deficient ob/ob but not in HFD-fed obese mice [290,291]. Additionally, it was suggested that AT-infiltrating macrophages contribute to the increased RBP4 levels, where differentiated macrophages were shown to produce RBP4 [292]. Nevertheless, the contribution of AT-derived RBP4 to the increased levels of circulating RBP4 in states of insulin resistance was contested [293,294]. Therefore, it is believed that AT-derived RBP4 play important paracrine and autocrine functions that are confined to the AT. Importantly, obese and diabetic individuals as well as GLUT4^−/−^ insulin-resistant mice exhibit elevated serum levels of RBP4 [295,296]. Moreover, transgenic mice overexpressing human RBP4, and mice injected with recombinant RBP4 exhibit insulin resistance, while the genetic deletion of RBP4 enhances insulin sensitivity in mice [295]. This could be partially explained by the fact that RBP4 induces the activation of JNK and TLR4 signaling pathways in AT-resident macrophages and thus contributes to AT inflammation-associated insulin resistance [297,298]. Indeed, HFD-fed RBP4^−/−^ mice exhibit reduced AT inflammation and insulin resistance in comparison to their wild type counterparts [298]. Intriguingly, adipocytic and macrophagic expression of RBP4 is strongly inhibited by TNF-α [292,299]. Similarly, IL-1β was shown to inhibit adipocyte RBP4 expression through binding to the IL-1βR and the downstream activation of NF-κB [300]. As RBP4 was shown to prime the NLRP3 inflammasome for IL-1β production through the TLR4/MD2 receptor complex and TLR2 in a glucose-dependent manner [301], it becomes plausible that IL-1β-mediated reduction of RBP4 expression is a compensatory mechanism that counteracts the development of further inflammation. Nevertheless, HFD-fed mice overexpressing RBP4 exhibited increased perigonadal AT expression of IL-1β [301].

The RBP4/STRA6 axis was also shown to partly modulate adipogenesis through mediating a bidirectional cellular retinol transport that controls RARα activity and the subsequent inhibition of adipocyte differentiation [302]. Moreover, RBP4 suppresses adipocyte differentiation through the inhibition of insulin signaling [303]. This possibly occurs as a result of the blocking of IRS1 phosphorylation at serine307 and the downstream phosphorylation of ERK1/2, independent of the autophosphorylation of the insulin receptor [304]. Nevertheless, STRA6 expression is reduced in perigonadal AT and in the stromovascular fraction of the subcutaneous and perigonadal AT in mice models of diet-induced and genetically induced obesity [305], which possibly limits the anti-adipogenic potential of the RBP4/STRA6 axis. Moreover, mice with the adipose-specific deletion of STRA6 exhibit enhanced leanness, reduced leptin and insulin levels, and enhanced WAT browning [305]. Consistently, adipocytes exposed to glucolipotoxicity and VAT tissue of HFD-fed insulin-resistant rats exhibit significantly increased RABP4/STRA6 expression that mediates AT inflammation and insulin resistance [306]. Serum and AT RBP4 levels were also suggested to contribute to systemic insulin resistance by enhancing adipocyte basal lipolysis and activating AT proinflammatory macrophages [307].

Importantly, RBP4 secretion by the PPAT has not yet been evaluated and the possible effects of RBP4 on PCa and in the context of metabolic dysfunction-associated prostatic neoplasm formation is largely uninvestigated. Although the precise mechanisms are poorly understood, RBP4 was suggested to modulate the crosstalk between AT dysfunction and the development of cancer [308]. Indeed, RBP4 is associated with the enhanced thrombogenicity of procoagulant states and knocking down RBP4 inhibits the growth of PC-3 cells in vitro [309,310,311]. Therefore, the possible roles of RBP4 at the intersection of these pathologies warrants further investigation.

### 8.8. FABP4

Fatty acid binding protein 4 (FABP4), a member of the cytoplasmic fatty acid binding protein multigene family is secreted by adipocytes and its expression increases in cases of obesity and the MetS [312,313,314,315,316]. Exogenous FABP4 is taken up by PCa cells, activates the PI3K/Akt pathway, and enhances PCa invasion [312]. Indeed, FABP4 inhibition in a mouse model of PCa reduced tumor growth and metastasis, partly by inducing prostatic epithelial cell DNA damage and apoptosis [312]. Furthermore, FABP4 knockdown provided evidence that FABP4 regulates PAR-1-mediated expression of IL-6 and VEGF in cultured adipocytes [317]. Moreover, IL-6 production was also partially suppressed by knocking down PAR-1 [317]. Alternatively, FABP4 is also expressed in PCa cells and FABP4 expression is associated with aggressive PCa phenotypes [318]. The high expression and secretion of FABP4 by PCa cells stimulate cellular invasiveness through PI3K and MAPK-mediated upregulation of MMPs [318]. Additionally, FABP4-enhanced cellular invasiveness is associated with an increased secretion of IL-8 and IL-6, which was abrogated by pharmacologically inhibiting FABP4 [318]. Moreover, lipid-supplying bone marrow adipocytes, via increasing metastatic PCa cells expression of FABP4, IL-1β, and HMOX-1, enhance metastatic PCa cell growth and invasiveness [319]. Mechanistically, it was suggested that FABP4 pathway bidirectionally interacts with that of PPARγ to drive aggressive tumor behavior in bones [319].

### 8.9. IL-6

Over three decades have passed since the identification of IL-6 and IL-6R expression in BPH, PCa, and PCa cells [320,321]. Indeed, high intracellular levels of IL-6 were detected in tissues of patients with localized PCa, which indicates an oncogenic function of IL-6 during early prostatic carcinogenesis. Indeed, it was demonstrated in a prostate-specific IL-6 transgenic mice that IL-6 autonomously induces prostate neoplasm secondary to the activation of STAT3/IGF signaling [322]. The expression of IL-6 is primarily governed by the activation of upstream inducers including IL-1, TNF, platelet-derived growth factor, members of the AP-1 complex NF-κB, and TGFβ [323]. On the other hand, IL-6 signals by utilizing the adaptor molecule gp130 via canonical membrane-bound IL-6R and/or soluble IL-6R trans-signaling in IL-6R-gp130^+^ cells to initiate the downstream activation cascade [322]. The binding of IL-6 to its receptor elicits the downstream activation of multiple signaling pathways including the JAK-STAT3 pathway, which promotes cell cycle progression, tumor invasiveness, and host immune-system evasion [324]. Importantly, activated STAT3 expression is widely observed in the majority of PCa patients, highlighting STAT3 as an oncogene [184]. Other pathways, including the ERK1/2-MAPK and the PI3-K pathways, allow IL-6 to promote PCa cell proliferation and inhibit apoptosis in vitro and in vivo [324]. Indeed, IL-6 is associated with aggressive PCa phenotypes and may be involved in the metastatic process via the regulation of epithelial-mesenchymal transition (EMT) and the homing of cancer cells to the bone [325]. This has led to the development of anti-IL-6 therapies that, despite the decreased phosphorylation of STAT3 and p44/p42 MAPK as well as suppressing IL-6 downstream signaling [326], failed to improve the survival of patients with metastatic PCa in clinical trials [327,328].

VAT exhibits a more pronounced secretion of IL-6 in comparison to SAT, and PPAT is suggested to be a valuable source of IL-6 in PCa patients, which correlates with cancer aggressiveness [155,186,329]. Intriguingly, prostate-specific IL-6 transgenic mice exhibit, in addition to autonomously induced IL-6/STAT3/IGF signaling axis-mediated prostate neoplasms, an amplification of PPAT inflammation [322]. Indeed, the incubation of periprostatic adipocytes with PC-3-conditioned media enhances IL-6 production [148], which signifies that an intimate paracrine communication between the PPAT and the prostate partly drives IL-6 secretion in PCa. Moreover, higher Gleason scores in PCa patients associate positively with PPAT STAT3 signaling and IL-6 secretion [184].

### 8.10. TNF-α

Furthermore, TNF-α is a multipotent proinflammatory cytokine belonging to the TNF/TNFR cytokine superfamily that is mainly produced by immune cells, with particularly activated macrophages during acute phase reaction or innate immune response [330,331]. However, other sources of TNF-α were shown to include white blood cells, mesangial cells, fibroblasts, astrocytes, Kupffer cells, smooth muscle cells, keratinocytes, tumor cells, and adipocytes [154]. Localized, low-level expression of TNF-α participates in beneficial tissue remodeling, regeneration, and repair [332]. In circulation and under normal conditions, TNF-α is either undetectable or found in low concentrations [333]. On the other hand, as an endogenous pyrogen, TNF-α is able to mediate fever, apoptotic cell death, sepsis (through IL-1 and IL-6 production), cachexia, tumor regression, as well as carbohydrate metabolism and adipogenesis inhibition [332,334].

On the molecular level, TNF-α exists either in a soluble form or as a non-soluble membrane-bound anchor transmembrane domain in the propeptide that is processed by a matrix metalloproteinase named TNF-α-converting enzyme (TACE) [330]. The signaling cascade downstream of TNFRI and TNFRII is complex, involving multiple adapter proteins, which are recruited upon the binding of TNF-α to its receptor, and regulate at least four distinct pathways, namely the a pro-apoptotic pathway that is induced by binding caspase-8 to FADD; an anti-apoptotic program that is activated by the binding of cellular inhibitor of apoptosis protein-1 (cIAP-1) to TRAF2; a pro-inflammatory pathway which is mediated through the activation of activating protein-1 (AP-1) through TRAF2 via a JNK-dependent kinase cascade; and NF-κB activation by RIP [335,336,337]. Proinflammatory activities performed by TNF-α also involve the generation of prostaglandin and the induction of COX-2 [338]. However, in chronic inflammatory responses, TNF-α activates various caspases which leads to the apoptosis of inflamed cells [339]. Moreover, in addition to an increase in circulating levels, TNF-α is overexpressed in the insulin-resistant AT of obese individuals and specifically in VAT rather than SAT [340], which is lessened following weight loss [341]. Specifically, insulin resistance in adipocytes induced by TNF-α and IL-6 attenuates the inhibitory effect of insulin on lipolysis and FFA release [342,343]. TNF-α also directly inhibits the adipocyte production of adiponectin [344], and enhances their production of resistin [345].TNF-α reduces insulin synthesis, glucose transporter type 4 expression, and the serine phosphorylation of insulin receptor substrate-1 [346]. As a result, increased TNF-α synthesis and secretion in the context of inflammation have been linked to the development of insulin resistance and the pathogenesis of metabolic disorders [339,347].

The diverse activities of TNF-α led to the simultaneous and paradoxical pursuit of TNF-α as an anti-tumor strategy. However, TNF-α was implicated in the induction of chemo-resistance, promoting invasion, and increasing the risk of metastasis in several cancers [331,333,336,348]. Moreover, TNF-α promotes both DNA damage and inhibits DNA repair by up-regulating nitric oxide (NO) (non-cGMP)-dependent pathways [349]. Clinically, elevated serum concentrations and increased expression of TNF-α are present in various pre-neoplastic and malignant conditions [350]. Indeed, TNF-α promotes MMP expression, invasiveness, and metastasis via NF-κB and STAT3, in addition to a direct effect causing genetic damage to cells. TNF-α also enhances malignant cell survival and induces EMT [351]. Other experiments performed in mice have shown that the neutralization of TNF-α can convert inflammation-promoted metastatic growth to inflammation-induced tumor regression, dependent on IFN-induced TRAIL expression [352].

TNF-α expression has been confirmed in the tumor micro-environment in several malignancies including PCa [148,331,353]. Interestingly, serum TNF-α has been reported as a biomarker for PCa diagnosis and significantly correlated with the aggressiveness of PCa [154]. Indeed, blood TNF-α concentrations were found to be higher in PCa patients with advanced, cachectic disease [354]. TNF-α was found to influence PCa progression, increase the risk of metastasis and mediate androgen independence [154,355]. Interestingly, TNF-α was shown to be one of the adipokines secreted in a paracrine manner from PPAT to influence PCa progression [154]. Indeed, PPAT TNFα and VEGF immunostaining positively correlates with the grade and stage of PCa in men undergoing radical prostatectomy for the clinically localized disease [154]. However, it was reported that TNF-α was secreted by PPAT explants stimulated with PC3 conditioned medium, but not by stromal vascular fractions that did not contain mature adipocytes [148,356]. In addition, TNF-α and VEGF secreted by activated mature PPAT adipocytes facilitate PPAT lipoblasts seeding to the PCa tumor stromal microenvironment, thus promoting PCa progression by inducing vascularity and increasing vascular permeability [154]. On the other hand, the stimulation of PPAT explants with PC3 culture media induced TNF-α upregulation along with other proinflammatory adipokines associated with cancer progression such as osteopontin and IL-6 and reduced the expression of the protective adipokine adiponectin [148].Adding to its numerous roles, PPAT TNF-α has a lipolytic role which increases the release of FFAs, a major source of energy for PCa [154,357].

### 8.11. Osteopontin

Osteopontin (OPN) is a widely expressed secreted glycosylated phosphoprotein encoded by the SPP1 gene that is involved in various pathophysiological processes including obesity, diabetes, and PCa [358,359]. Circulating OPN levels increase in states of metabolic dysfunction such as obesity, diabetes, and NAFLD, and is thought to mediate, at least in part, their associated cardiovascular complications [358]. It was also suggested that the MMP9-mediated cleavage of OPN enhances OPN detrimental activities in the AT in obesity [360]. Moreover, the development of obesity and insulin resistance is associated with local AT augmentation of OPN expression with the particular upregulation in AT-resident macrophages in isolation of increased systemically circulating OPN [361,362].

Indeed, the specific silencing of osteopontin in epidydimal AT macrophages of obese mice significantly improved glucose tolerance [363]. The significance of OPN upregulation in the process of AT inflammation is evident by the virtual absence of AT macrophages in SPP1^−/−^ mice [364]. Moreover, HFD-fed OPN-deficient mice exhibit improved insulin resistance, which is associated with reduced AT macrophage infiltration, reflecting impaired macrophage motility and attenuated monocyte recruitment [365]. OPN-deficient mice also exhibit reduced AT ECM remodeling evidenced by decreased fibrosis, a reduced activity of MMP2 and MMP9, and lower expression of collagen and TGF-β1 [366]. Similarly, OPN neutralizing antibody-treated, HFD-fed mice exhibit improved insulin sensitivity and attenuated AT macrophage infiltration and inflammation [367]. This was associated with an increased macrophage apoptosis and significantly reduced JNK activation [367]. Nevertheless, although SPP1^−/−^ mice fed a HFD exhibit lower AT macrophage infiltration in comparison to their control counterparts, the M1/M2 ratio in these mice is higher, which signifies that OPN might not be required for M1 macrophage polarization, but is instead required for the induction of phagocytosis [368]. OPN elicits its functions partly through its binding to the multifunctional membrane receptor CD44, for which circulating levels and expression in the AT increases in states of obesity and insulin resistance [369,370]. Moreover, CD44 density on AT macrophages is associated with proinflammatory M1 polarization [369]. This is supported by the observation that CD44-deficient mice exhibit reduced susceptibility to HFD-induced AT macrophage infiltration, AT inflammation, and insulin resistance [371].

Osteopontin induces brown adipogenesis in cultured preadipocytes through the activation of the PI3K/Akt pathway in a CD44-dependent manner and the downregulation of OPN in WAT of mice exacerbates obesity and inhibits WAT browning via the inhibition of PPARγ-mediated activation of PI3K/Akt signaling [372,373,374]. Nevertheless, another study suggested that OPN-deficient mice exhibit higher body temperature and enhanced BAT function [366].

As the PPAT possibly has the potential to modulate the PCa cell microenvironment through the activation of MMPs [183], it becomes plausible that these effects occur downstream of an increased PPAT secretion of osteopontin. Indeed, PC-3-conditioned media-stimulated PPAT explants exhibit an increased secretion of osteopontin, which is associated with increased MMP-9 activity [148]. Moreover, increased OPN secretion was observed in the AT of adipocyte-specific p62/SQSTM1 deletion murine model of PCa [375]. The induction of OPN resulted in enhanced prostate tumor cell fatty acid oxidation and invasion, leading to aggressive metastasis suggesting that the AT metabolism directly modulates PCa.

Despite conflicting available literature on the utility of osteopontin (OPN) plasma levels as a marker for the detection of PCa or high-grade PCa, OPN represents a promising diagnostic and prognostic biomarker for PCa patients [376]. Osteopontin is expressed in PCa cells, where alternative splicing gives rise to three splicing isoforms, namely OPNa, OPNb, and OPNc, of which OPNb and OPNc were shown to have pronounced pro-tumorigenic effects in androgen-responsive and androgen-independent PCa [377]. PCa patients present significantly higher levels of the three splice variants in comparison to patients with BPH and patients with symptomatic BPH express higher levels of OPN in comparison to patients with incidental BPH [378,379]. Indeed, OPN progressively augmented expression stage-dependently associates with prostate PCa [380]. OPN was suggested to differentially potentiate LNCaP cells proliferation, invasiveness, and intravasive potential [380]. Particularly, OPNc was suggested to promote cancer progression where it activates AR signaling in LNCaP cells in a PI3K signaling-dependent pathway [381]. Moreover, conditioned media from OPNc-overexpressing PC-3 cells was shown to enhance endothelial cell adhesion, proliferation, and migration [382]. Of particular interest, OPN possesses a functional arginine-glycine-aspartic acid (RGD) domain that allows its binding to integrins and the OPN/α_v_β_3_ axis was shown to enhance VEGF expression downstream of ERK1/2 phosphorylation [383]. The OPN/α_v_β_3_ axis was also shown to enhance PC-3 cell chemotaxis and chemo-invasion by means of the upregulation of plasminogen activators [384].

Both MMP3 and MMP7 were shown to cleave OPN, which upon cleavage exhibits potentiated adhesive and migratory stimulation potential [385]. Moreover, OPN-mediated Akt activation occurs either by the integrin α_v_β_3_ or the novel splice variants of the cell surface receptor CD44 associated with PCa [386,387], which results in increased c-Raf phosphorylation and downstream phosphorylation of ERK1/2. Subsequent to OPN-induced Akt activation, GDK-3β, a regulator of β-catenin is inactivated, which leads to the nuclear translocation of β-catenin and the increased expression of MMP7 and CD44, two known TCF/LEF transcription factors [388]. Indeed, the intracellular domain of CD44 represents a co-transcription factor for RUNX2 that regulates MMP9 expression in PC-3 cells, which contributes to cellular migration, invasion, and tumorsphere formation [389,390]. In fact, RUNX2 is abnormally expressed in metastatic PCa cells where it facilitates tumor growth and osteolysis possibly through a RUNX2/Smad signaling axis [390,391]. As such, RUNX2 downregulation was shown to inhibit tumor growth and bone metastasis in Pca [392]. This is of particular interest as the activation of the OPN/MMP9 pathway was shown to correlate with PCa progression [393].

OPN downregulation in PC-3 cells induces pronounced cell cycle arrest, apoptosis, and a reduced colony formation potential [394]. Moreover, OPN induces the expression of MMP-2 and MMP-9 through the activation of IKK-2/NF-κB signaling pathway and inhibiting OPN expression in PC-3 cells downregulates the expression of MMP-2 and MMP-9 [395]. Similarly, OPN-suppressed DU145 cells exhibit reduced growth and invasiveness in vitro and in mouse xenograft models [396]. Moreover, PC-3 cells overexpressing OPNb or OPNc exhibit pronounced resistance to docetaxel-induced cell death via enhanced EMT in comparison to control and OPNa-overexpressing PC-3 cells [397]. OPN was also shown to upregulate P-glycoprotein (P-gp) expression in PC-3 cells by binding the integrin α_v_β_3_ and thus confer chemotherapeutic resistance, while the knocking down of OPN enhanced cell death in response to other P-gp substrates including paclitaxel, doxorubicin, actinomycin-D, and rapamycin [398].

Intriguingly, OPN inflammatory activity could be modulated by thrombin-mediated proteolytic cleavage at a highly conserved cleavage sites, which likely occurs in the tumor microenvironment. This cleavage exposes an integrin-binding motif that is thought to promote OPN biological activities [399]. Deletion of the thrombin-cleavable site of OPN in breast cancer cells resulted in reduced in vitro cellular adhesion and enhanced in vivo tumor growth and lymph node metastasis [400]. Moreover, it is suggested that thrombin inhibition can reduce the malignant and metastatic behavior of breast cancer cells in an OPN-dependent and OPN-independent manners [401]. Nevertheless, such activities of thrombin-cleaved OPN have not been investigated in the context of prostate cancer.

### 8.12. Chemerin

Chemerin (retinoic acid receptor 2, RARRES2) is an endogenous leukocyte chemoattractant that, through its cognate G protein-coupled receptor CMKLR1, participates in the early stages of acute inflammation [402]. Although states of metabolic dysfunction including insulin resistance, obesity, and T2D are associated with increased levels of circulating chemerin and CMKLR1 expression in SAT and VAT [402,403,404,405,406,407,408,409], there remain controversies concerning whether chemerin upregulation participates in disease pathogenesis or represents a compensatory mechanism. Particularly, and prior to the proteolytic cleavage of its C-terminal by various proteases including plasmin, FXIa, and FXIIa, circulating prochemerin is biologically inert [410]. Chemerin processing seems to be altered in states of metabolic dysfunction, however, the respective proteases involved therein are not fully characterized. Moreover, a second poorly studied receptor of chemerin is the G protein-coupled receptor GPR1, which binds chemerin with comparable affinity to CMKLR1. Although it is highly expressed in BAT and WAT, GPR1 expression dominates in the stromovascular fraction of WAT [411]. Of note, it was shown that GPR1 but not CMKLR1 bioactivity positively correlates with AT inflammation [412]. Moreover, HFD-fed homozygous and heterozygous GPR1^−/−^ mice exhibit more pronounced glucose intolerance in comparison to their wild type counterparts in isolation of changes in body weight, adiposity, or energy expenditure [411]. Therefore, a deeper investigation of chemerin isoform specific upregulation in the context of metabolic diseases and receptor bioactivity is required. Chemerin production in the AT is enhanced by adipocyte hypertrophy and low grade inflammation [413,414], as active chemerin expression is enhanced in adipocytes following TNF-α treatment in vitro and in vivo possibly through elastase and tryptase-mediated proteolytic cleavage [404,415,416]. Consistently, leptin and leptin receptor deficient mice, which have an elevated AT expression of TNF-α, display elevated serum chemerin levels [415]. Similarly, IL-1β significantly induces chemerin secretion in cultured adipocytes possibly by activating JNK2, NF-κB, p44/42 MAPK, and PI3K signaling pathways [417].

Owing to the differential expression of chemerin across adipose depots in obesity, chemerin enhances adipogenesis in a depot-specific manner [418,419]. Indeed, the expression of chemerin and CMKLR1 increases during adipocyte differentiation and is suggested to induce ERK1/2 phosphorylation and lipolysis in differentiated adipocytes [420]. As such, knocking down CMKLR1 impairs 3T3-L1 adipocytic differentiation and alters the metabolic function of mature adipocytes [421]. Moreover, chemerin/CMKLR1 signaling plays an important role in clonal expansion during adipocyte differentiation downstream of PPARγ activation [422].

Whereas obesogenic diets were shown to enhance chemerin expression in BAT and cold acclimation was shown to diminish chemerin expression, with both occurring independent of adrenergic control [423]. Chemerin signaling was also suggested to suppress AT browning and thermogenesis through modulating type 2 innate immunity. Indeed, the expression levels of chemerin and CMKLR1 decrease in mice inguinal WAT following cold exposure, and the deletion of either adipocytic chemerin or CMKLR1 enhances cold-induced thermogenesis through potentiating IL-33 secretion and type 2 innate immunity [424]. Mechanistically, IL-33 expression is suppressed by the chemerin/CMKLR1 axis by dampening cAMP/PKA signaling. Importantly, genetically blocking adipocytic CMKLR1 protects mice against DIO [424]. Contrastingly, chemerin is suggested to play a role in brown adipocyte differentiation as chemerin knockdown decreases brown adipocyte differentiation [425]. Moreover, CMKLR1^−/−^ mice exhibit reduced propensity for cold-induced thermogenesis, and CMKLR1 deficiency suppresses the expression of thermogenic genes in stromovascular fibroblast-derived adipocytes in vitro [426]. Nevertheless, HFD-fed chemerin^−/−^ mice were prone to diet-induced AT expansion, obesity, and insulin resistance [425]. Moreover, although chemerin^−/−^ mice exhibit larger BAT, chemerin-deficient brown adipocytes exhibit lower mitochondrial content and impaired mitochondrial function, which are associated with reduced energy expenditure [425]. Although exogenous chemerin was shown to exacerbate glucose uptake in obese and diabetic mice, which exhibit altered WAT, skeletal muscle, and hepatic expression of CMKLR1 and GPR1 [427]. Nevertheless, several reports highlighted a positive role for chemerin in regulating glucose homeostasis where chemerin^−/−^ mice exhibit glucose intolerance and insulin resistance when placed on either an HFD or control diet [428]. This was associated with the reduction of AT and the skeletal muscle expression of GLUT4 and PGC-1α. Moreover, it was shown that HFD-fed CMKLR1 knockout and heterozygous mice exhibit a higher tendency to develop obesity and impaired glucose homeostasis [426,429]. The chemerin treatment of rat and mouse-derived epidydimal as well as 3T3-L1 adipocytes enhances insulin sensitivity and insulin-induced glucose uptake, a phenotype that was abrogated following CMKLR1 deletion [429,430]. Another study highlighted that CMKLR1^−/−^ mice, either fed a control or a HFD, despite displaying a lower percentage of body fat and decreased AT inflammation, were glucose intolerant [431]. Moreover, CMKLR1^−/−^ mice did not show a higher propensity to develop obesity when challenged with an HFD, which was also associated with a lack of adipose inflammation, or adipocyte dysfunction [432].

Targeting chemerin and its receptor CMKLR1 has emerged as a valuable therapeutic approach against insulin resistance, T2D, and cancer [433]. One study suggested that chemerin circulating levels positively correlates with the Gleason score in PCa patients [434]. Nevertheless, chemerin levels could neither discriminate between patients with PCa and BPH, nor were chemerin levels increased in obese patients with clinically localized PCa in comparison to their non-obese counterparts [434,435]. This contradicts the idea of AT-derived chemerin playing a role in PCa. Nevertheless, while serum chemerin levels paralleled the increase in the Gleason score, the opposite trend was observed in the prostate tumor tissue [436]. Moreover, chemerin expression is downregulated in PCa tissues in comparison to those of BPH, with a greater reduction observed in castration-resistant PCa [437]. Although our understanding of the roles played by chemerin in the context of PCa are fairly limited, it was shown that human tumors exposed to exogenous chemerin upregulate PTEN expression with the concomitant suppression of PD-L1 expression, possibly by the activation of the PI3K/Akt/mTOR signaling pathway, a phenotype that was abrogated following CMKLR1 knockdown [438]. It was also shown that chemerin treatment reduces tumor migration and enhances T cell-mediated tumor lysis, which suggests a role for chemerin in improving T cell-mediated immunotherapies.

### 8.13. Apelin

Several studies highlighted that increased serum levels of apelin are associated with increased adiposity, obesity, insulin resistance, MetS, and diabetes [439,440,441,442]. Apelin is secreted by white and brown adipocytes and the expression level of apelin increases during adipocyte differentiation in an insulin-dependent manner, which is associated with the stimulation of PI3K, PKC, and MAPK [440,441,443]. In fact, defective apelin signaling is associated with altered lipid metabolism in the AT of HFD-fed mice, improved glucose tolerance, reduced serum insulin levels, and enhanced insulin sensitivity while the administration of modified analogues ameliorates several aspects of metabolic dysfunction [444,445,446,447]. Pertinent to the inflammatory and hypoxic natures of AT in states of metabolic diseases, cultured hypoxic adipocytes exhibit an increased expression of apelin and leptin and reduced expression of adiponectin in response to an increased expression of HIF-1α [448]. Moreover, TNF-α enhances apelin expression in adipocytes in vivo and in vitro via certain pathways dependent on the activation of PI3K, JNK, and MAPK [449]. In human AT, apelin dose-dependently stimulates AMPK phosphorylation and the subsequent uptake of glucose [443]. However, apelin was shown not to have a lipolysis-promoting role in isolated human adipocytes or AT explants [443]. In fact, apelin reduced lipolysis in cultured adipocytes by increasing HSL phosphorylation and AMPK activation-mediated acetyl CoA carboxylase phosphorylation [450]. Apelin binds its receptor APJ and was shown to both centrally and peripherally regulate glucose homeostasis and energy expenditure either through NO-dependent mechanisms, or via an increasing vascular mass and muscular mitochondrial biogenesis, respectively [451,452]. Moreover, an acute intravenous injection of apelin enhanced glucose utilization in AT and skeletal muscle which involves eNOS, AMPK, and Akt signaling [453]. Indeed, intracerebroventricular administration of apelin in rodents reduces food intake [454]. However, chronic intracerebroventricular infusion of apelin in mice increases the expression of pro-inflammatory markers in the hypothalamus in addition to increasing serum IL-1β levels [455]. These mice exhibited a significantly lower energy expenditure that was associated with a decreased expression of PGC-1α, PRDM16, and UCP1 expression in BAT [455]. However, cAMP, an upstream regulator of PGC-1α, either cold-exposure-induced or pharmacologically induced, was shown to enhance apelin expression in adipocytes in vitro and in vivo [456]. Intraperitoneal administration of apelin to DIO mice resulted in a significant reduction of body adiposity, serum insulin, and serum triglyceride levels, which was associated with increased serum levels of adiponectin and reduced serum levels of leptin [457]. Importantly, apelin administration enhances BAT thermogenesis, evidenced by an upregulation of UCP1, and an increase in body temperature and oxygen consumption rate [457]. Additionally, intracerebroventricular administration of apelin enhanced the locomotor activity in mice in addition to body temperature and weight gain [458]. Central administration of apelin in high doses in fasted mice provokes hyperinsulinemia, glucose intolerance and insulin resistance [452]. Nevertheless, apelin paraventricular injections were shown to enhance sympathetic innervation into BAT and may therefore, increase its thermogenic activity [459].

The apelin/APJ axis plays pivotal roles in the development of tumors via the promotion of cellular proliferation, angiogenesis, metastasis, and the development of drug resistance [460,461]. Particularly, apelin upregulation occurs more frequently in patients with advanced stage PCa and metastasis in which miR-224 is downregulated [462]. Furthermore, miR-224 was shown to directly target apelin transcripts and the forced expression of miR-224 inhibits PCa cell invasion and migration through the suppression of apelin expression [462]. It was therefore postulated that the synergistic downregulation of miR-224 and upregulation of apelin may predict the biochemical recurrence-free survival in PCa patients.

### 8.14. C–C Motif Chemokines

#### 8.14.1. CCL2

As is known, CCL2, also known as monocyte chemoattractant protein-1 (MCP-1), is primarily produced in the stromovascular fraction of PPAT, and its expression increases in obesity and correlates with decreased survival in patients with PCa [463]. In addition to monocyte recruitment into AT, CCL2 and its receptor, CCR2, form an axis that has been identified as a cornerstone in recruiting regulatory T cells in a sex hormone-dependent manner in inflamed VAT [464]. Certainly, CCL2 has been shown to play a unique role among several cytokines that influence the recruitment of AT macrophages, and the link between AT inflammation and insulin resistance [465,466,467]. Indeed, a change in the secretory pattern of adipocytes due to inflammation and the subsequent modifications in its cellular composition due to excess visceral adiposity around the prostate creates an inflammatory microenvironment that nurtures PCa [187,468].

As such, at the molecular level and as expected, upstream of CCL2 expression are inflammatory stimuli, including IL-1, IL-4, IL-6, TNFα, TGFβ, LPS, and IFNγ [469,470,471,472,473,474]. On the other hand, downstream and upon binding to CCR2, CCL2 activates several signaling pathways, including those of JAK2/STAT3 [475], MAPK [476], and PI3K [477]. Significantly, upon the activation of CCL2/CCR2, the PI3K/Akt pathway activates mTORC1 and up-regulates survivin which is a key molecule protecting PCa cells from autophagic death [478]. Accordingly, CCL2 is overexpressed in primary prostatic tumors and is mirrored by high expression levels of its CCR2, that is associated with a higher Gleason score and clinical pathologic stages [479,480].

Interestingly, PPAT-derived CCL2 induces PCa growth, progression, cell migration, invasion, and metastasis [481,482]. Furthermore, the mRNA and protein expression of CCR2 are higher in aggressive cell lines such as DU145, PC-3, and C4-2B compared with levels in androgen-sensitive LNCaP cells and non-neoplastic prostate epithelial cells [480,482]. Functionally, CCL2 is involved in modulating tumor cell growth by regulating the infiltration of macrophages into tumors, promoting osteoclast maturation, and suppressing cytotoxic lymphocytes in PCa [483,484,485]. Moreover, upon the induction of CCL2 and consequently activating the downstream ERK signaling pathway, WNT5A induces castration-resistant PCa by mediating macrophage infiltration [486]. Additionally, in advanced PCa, CCL2 expression is notably higher in the metastatic tumor–bone microenvironment compared to that in bone-marrow adjacent to the tumor [487]. Mediated by CCL2 stimulation of PCa cells, bone metastasis is dependent on the upregulated expression of the actin-associated protein, PCNT1, and on Rac activation [488]. Importantly, CCL2 has been implicated as one of the main cytokines involved in tumor cell re-establishment in the bone marrow in PCa [489]. Moreover, knocking down CCR2 abrogated PCa invasiveness to the bone, whereas, in an in vivo model of PCa metastasis, inhibiting CCL2 activity with neutralizing antibodies decreased the overall tumor burden [490,491].

Additionally, CCL2 increases prostate cancer cell migration through the CCR2 receptor involving the upregulation of αvβ3 integrin [489]. This highlights the role of the CCL2/CCR2 axis in the tumor microenvironment in stimulating PCa expansion and metastasis [492,493]. Moreover, the induction of adiponectin by means of the PPARγ agonism suppresses the elevation in CCL2 levels by blocking TNF- α signaling, thereby attenuating obesity-associated PPAT inflammation [103]. Furthermore, due to the increased secretion of CCL2 by adipocytes, a high infiltration of macrophages is observed in the PPAT of obese animal models [104]. Incidentally, when macrophages are recruited by adipocytes in the tumor microenvironment, they are metabolically reprogrammed, become M2 polarized, and promote tumor growth and progression [494,495]. Consequently, Ccr2-deficient animals have significantly fewer AT macrophages than wild-type mice, and the genetic deletion or pharmacological inhibition of Ccr2 reduce AT macrophage content and improve AT inflammation [496,497]. Moreover, obesity-associated PPAT inflammation, characterized by CLS formation, is associated with high-grade PCa. Treatment with pioglitazone, a PPARγ agonist, reduces CLS density in PPAT and suppresses CCL2 levels and the levels of two of its upstream mediators, TNF-α and TGF-β [103,469,470,471,472,473,474].

#### 8.14.2. CCL7

Furthermore, CCL7, also known as monocyte chemotactic protein 3 (MCP-3), is expressed by various types of cells under physiological conditions and by tumor cells, leukocytes, endothelial cells, and fibroblasts under pathological conditions [498,499]. Although CCR1, CCR2, CCR3, and CCR5 are widely recognized as the main functional receptors of CCL7, PCa is mainly dependent on the CCR3/CCL7 signaling axis, which contributes to the tumor microenvironment [500]. Notably, CCL7 and its respective receptor CCR3, are upregulated in AT in human obesity and are associated with increased inflammation [501]. Furthermore, CCL7 is a potent chemoattractant for a variety of leukocytes, including monocytes, eosinophils, basophils, dendritic cells (DCs), natural killer (NK) cells, and activated T lymphocytes [502]. Furthermore, both EGF and TNF-α were shown to induce the expression of CCL7 in adipocytes [503]. Moreover, PC3 cells were shown to secrete pro-metastatic factors, including CCL7 and TGF-β, which accelerate PCa growth [504].

Adipocytes of the PPAT were shown to support the directed migration of PCa cells via the secretion of CCL7, which diffuses to the peripheral prostatic regions, stimulating the migratory capacity of CCR3-expressing cancer cells [481]. In the obese state, adipocyte secretion of CCL7 is increased, which facilitates extraprostatic extension through the CCR3/CCL7 axis [481]. Importantly, the expression of CCR3 is associated with a more aggressive disease and a higher risk of biochemical recurrence. Upon inhibiting the CCR3/CCL7 axis, the observed obesity-potentiated tumor cell migration and invasion are nullified [481]. Indeed, it was recently demonstrated that soluble factors, including CCL7, released by human primary bone-marrow adipocytes can support the directed migration of PCa cells in a CCR3-dependent manner. These effects are amplified by obesity and aging, two clinical conditions known to promote aggressive and metastatic PCa [505].

#### 8.14.3. CXCL12

Additionally, known as stromal-cell derived factor 1 (SDF-1), CXCL12 is another chemokine that is expressed by PPAT stroma and becomes increased in obesity, positively correlating with a decreased survival in patients with PCa [149]. Furthermore, CXCL12 expression is remarkably increased in the stromal vascular fraction of the PPAT and the stromal compartment of the ventral prostatic tissue of obese mice with myc-induced PCa [506]. This was accompanied by an increased expression of CXCR4 and CXCR7, two receptors of CXCL12, in the epithelial compartment of the prostate. It is noteworthy to mention that CXCL12 has been described as a novel, non-steroidal growth factor that promotes the growth of prostate epithelial cells via androgen receptor-dependent mechanisms in the absence of steroid hormones, supporting the therapeutic targeting of the CXCL12/CXCR4 axis as supplementary to targeting the androgen/AR axis to effectively treat castration resistant/recurrent PCa [507]. Moreover, CXCL12 was identified as one of 38 other cytokines whose levels differed between PCa patients and healthy controls, suggesting that CXCL12 may serve as a biomarker for the early diagnosis and prognosis of PCa [508].

At the molecular level, CXCL12 activates STAT3, NF-κB, and MAPK signaling and stimulates the migration of HiMyc PCa cells in vitro in a CXCR7/CXCR4-dependent manner [506]. Downstream of the activated CXCL12/CXCR4 axis, is the PI3K-Akt cascade which mediates CXCL12-stimulated migration and invasion [509]. Moreover, the CXCL12/CXCR4 axis promotes EMT, involving tumor progression locus 2 (Tpl2) kinase and activating the ERK1/2 pathway provoking progression and metastasis of castration-resistant PCa [510,511]. The increased expression of the CXCR4 protein is significantly associated with lymph node or bone metastasis, and Chen et al. suggested the use of CXCR4 for the noninvasive monitoring of PCa progression [512]. On the other hand, Pim kinases, a family of oncogenic kinases, was found to be likely to promote metastatic PCa growth by employing the CXCL12/CXCR4 pathway [513]. Fittingly, CXCL12 methylation downregulates tumor intrinsic CXCL12 protein expression, disrupting cellular feedback mechanisms to internalize membranous CXCR4 in PCa, thereby fostering metastasis [514], which is suggestive of the possible therapeutic potential of CXCL12 inhibitors [515]. In that context, several inhibitors of the CXCR4/CXCL12 axis and different effectors upstream and downstream the CXCR4/CXCL12 signaling pathway resulted in an inhibition of PCa growth, chemo-sensitization and suppression of EMT, migration and bone metastasis [516,517,518,519,520,521,522].

The CXCL12/CXCR4 pathway is also implicated in the activation of CD44^+^/CD133^+^ prostate progenitor population, a drug-resistant population of cells that lead to tumor relapse and affects differentiation potential, cell adhesion, clonal growth and tumorigenicity of PCa cells [523]. Alternatively, the enhanced expression of cytosolic superoxide dismutase (SOD1) that interacts directly with the first intracellular loop (ICL1) of CXCR4 and regulates CXCL12/CXCR4-mediated AKT activation leads to apoptosis and cell migration in PCa cells under hypoxic conditions when SOD1 is present [524]. On the other hand, the loss of the tumor suppressor PTEN leads to the subsequent phosphorylation of Akt, and the regulation of the CXCL12/CXCR4 signaling axis in PCa growth and bone metastasis [525]. Indeed, PTEN has been shown to mediate pro-proliferative signaling downstream of CXCL12/CXCR4 axis through the Raf/MEK/Erk or PI3K/Akt pathways [526]. Moreover, recent data suggest that the CXCL12/CXCR4 axis is functionally linked to the PD-1/PD-L1 immune checkpoint, where patients with high PD-L1 expression and aberrant CXCL12 methylation, presented with significantly shorter biochemical recurrence-free survival intervals than patients with either low PD-L1 expression or high PD-L1 expression plus normal CXCL12 methylation [527], suggesting the use of CXCL12 as a tool for predicting responsiveness to therapeutic interventions.

The deletion of the CXCL12 gene specifically in Pdgfr^+^ adipose stromal cells suppressed tumor growth and EMT, indicating that adipose stromal cells represent the main source of CXCL12 [149]. Indeed, obesity-induced EMT in prostate tumors of HiMyc mice was suppressed following the pharmacological depletion of adipose stromal cells, the key source of CXCL12 [149]. Importantly, CXCL12 expression in PPAT stroma increases in cases of obesity and correlates with a decreased survival of patients with PCa [149]. CXCL12 has been dubbed as an insulin-desensitizing factor in adipocytes, and was overexpressed in both fasting and obese AT. Fittingly, exogenously added CXCL12 induces a decrease in insulin-mediated signaling and glucose uptake, by phosphorylating and degrading IRS-1 in adipocytes [528]. On the other hand, adipocyte-specific ablation of CXCL12 enhanced AT and whole body insulin sensitivity [528].

Furthermore, CXCL12 modulates AT immune cells, promoting AT inflammation and subsequent obesity-associated insulin resistance. Indeed, CXCL12 is a known attractant molecule for T-cells, likely mediating the recruitment of T-lymphocytes into the AT [529]. Moreover, dose-dependent neutralization therapy of one of CXCL12 receptors, CXCR7, blocks CXCR7-mediated AT macrophages chemotaxis and ameliorates insulin resistance and inflammation in obesity [530]. High expression levels of CXCL12, as well as VEGF are correlated with lymph node metastatic prostate carcinoma compared to non-lymph-node metastatic cancer and are associated with poor cancer-specific survival after radical prostatectomy [531]. Notably, in castration-resistant PCa patient specimens, both the macrophage migration inhibitory factor (MIF) and CXCR7 are overexpressed, and their subsequent blockade inhibits castration-resistant PCa tumor growth and potentially prevents metastasis. However, CXCR7 was identified as a decoy receptor in the migration of androgen-responsive LNCaP cells toward CXCL12 [532]. Moreover, it was shown that in comparison to CXCR4, CXCR7 more prominently contributes to PCa cell migration as CXCR7-depleted PC-3 cells grow significantly slower in vitro and much slower in vivo, indicating CXCR7 may indeed control PCa cell proliferation in vivo [533]. Indeed, the CXCR4-mediated proliferation and metastasis of tumor cells was shown to be regulated by CXCR7 through its scavenging of CXCL12 [534]. Moreover, mutant p53 exerts a gain-of-function effect on CXCL12 and CXCL1 expression in fibroblasts, which may contribute to their ability to augment tumor growth. Suitably, the knocking down of CXCL12 strongly attenuates the positive impact of p53-deficient mouse embryonic fibroblasts on PC3 tumor growth [535]. In addition, the enhanced survival of docetaxel-treated PCa cells was mainly mediated by CXCR4 activation from the increased secretion of CXCL12 from CSF-1-activated tumor-associated macrophages [536]. Interestingly, PCa peripheral tumor-associated circulating natural killer cells acquire proinflammatory properties related to endothelial cell angiogenesis by increasing the expression of CXCL8 [537]. Moreover, CXCR6 signaling stimulates the conversion of mesenchymal stem cells into cancer-associated fibroblasts, which secrete CXCL12. The, CXCL12 expressed by cancer-associated fibroblasts binds to CXCR4 on tumor cells and induces EMT, which ultimately promotes metastasis [538].

## 9. Interventions Mitigating Periprostatic Adipose Tissue Inflammation

As PPAT inflammation is associated with an increased incidence of high-grade PCa and worse prognosis in PCa patients [129], interventions geared towards the suppression of PPAT inflammation are thought to improve such outcomes.

### 9.1. Caloric Restriction, Weight Loss, and Surgery

Intermittent fasting as well as caloric restriction regimens have long been shown to ameliorate the coagulatory and inflammatory status in the AT of metabolically impaired individuals and in rodent models. In addition to their inhibitory effect on AT cytokine release and immune cell recruitment [11], these interventions have been shown to halt AT thromboinflammatory processes. Several reports have documented that weight loss programs, either through the adoption of particular calorie-deficit diets or through surgery in overweight, obese, and morbidly obese individuals corrects not only dysregulated metabolic parameters, but also substantially ameliorates hemostasis [58,539,540]. This is evidenced by a reduction in the levels of fibrinogen, TF, and thrombin and their potential to drive thromboinflammation. Indeed, caloric restriction and drastic weight loss in obese mice result in marked reduction of SAT and VAT mass, adipocyte hypertrophy, as well as serum levels of cholesterol, triglycerides, leptin, PAI-1, FVII, and FVIII and increase in adiponectin serum levels [62]. Importantly, this was associated with reduced oxidative stress and IL-6 expression in the AT, and a higher expression of the antioxidant enzymes, catalase, SOD1, and glutathione peroxidase 1 [62]. Indeed, caloric restriction in obese mice was effective in reducing PPAT inflammation and the production of proinflammatory cytokines [130]. Nevertheless, the adoption of caloric restriction as a preventive measure of metabolic impairment remains controversial as caloric restriction was shown to distinctly affect AT cytokine and angiogenic factors secretion profile in obese and lean mice as it promotes a proinflammatory and a procoagulant state in the latter [541]. Moreover, thrombin generation was shown to be reduced in morbidly obese patients two years following bariatric surgery, which correlated with decreased cholesterol, triglycerides, and HbA1c and was inversely correlated with insulin resistance [57]. This is substantiated by the fact that a three-week duration of low caloric diet consumption in conjugation to balneological treatment in morbidly obese patients did not reduce the present hypercoagulable state, which argues that a significant fat mass must be reduced before the reestablishment of appropriate hemostatic parameters [542]. Laparoscopic and bariatric surgeries in obese and morbidly obese individuals have been associated with reduced endogenous thrombin potential (ETP) [543,544]. Additionally, endoscopic balloon placement in obese subjects mitigated their hypercoagulable state, as evidenced by a reduction in ETP, which persisted beyond balloon removal secondary to weight loss [545].

### 9.2. Estrogen Supplementation

The supplementation of 17β-estradiol or diethylstilbestrol, a synthetic estrogen, to DIO mice suppresses PPAT inflammation and downregulates its expression of CD68, MCP1, and TNF-α [130]. Although these effects are primarily attributed to a reduction in food intake and to a marked weight loss [130], this does not exclude the possible direct effects of estrogen on the PPAT.

### 9.3. Antidiabetic Drugs

Antidiabetic agents, such as metformin and thiazolidinediones, were shown to suppress PPAT inflammation, alter the development of PCa, and halt its progression. The pioglitazone, a PPARγ agonist, treatment in obese mice reduced CLS density in the PPAT and suppressed the expression of TNF-α, TGF-β, and MCP-1,effects which were abrogated in MCP1 KO mice [103]. Importantly, pioglitazone induced the expression of adiponectin and its receptor AdipoR2 in cultured 3T3-L1 cells which blocked TNF-α-mediated induction of MCP-1 [103]. Metformin was also shown to inhibit PCa growth in non-obese PC3 cells-inoculated HFD-fed immunosuppressed mice [546]. Metformin potentially acted via the modulation of the local expression of GH/IGF1 axis components, regulating tumor-associated processes such as apoptosis, necrosis and the cell cycle [546]. Metformin also inhibited cellular proliferation, migration, and prostate-specific antigen secretion from different PCa cell lines in vitro [546]. Nevertheless, metformin-mediated inhibition of PCa growth in vivo were studied in isolation of its potential effects on the PPAT.

## 10. Periprostatic Adipose Tissue: Novel Therapeutic Targets

### 10.1. Thrombin, Factor Xa, and PARs

Given the potential role of thrombin and FXa in driving PPAT inflammation, an essential instigator of the early stages of PCa development, it becomes evident that targeting thrombin and FXa either via direct oral anticoagulants or novel molecules developed to halt the proteolytic activities of thrombin and FXa with minimal bleeding risk, offer a valuable therapeutic intervention in the context of de novo malignancy prevention [547]. Particularly, several reports have elucidated the pleiotropic effects of DOACs extending beyond their perceived role of clotting reduction [26,548]. Possible effects of pharmacological agents such as rivaroxaban and dabigatran warrant further investigation, as thromboinflammatory processes are of particular importance in the context of metabolic dysfunction and PCa development. Some of the possible effects are highlighted in Figure 2. Indeed, the in vitro treatment of human-derived peripheral blood mononuclear cells with dabigatran significantly reduces their production of growth factors and chemokines in a dose-dependent manner [549]. Nevertheless, an extensive understanding of thrombin and FXa-mediated processes either promoting or inhibiting cancer development and progression is warranted. For example, PAR-1 deficiency in TRAMP mice results in the development of larger and more aggressive prostate tumors, partly due to a loss in the PAR-1-induced apoptosis of transformed epithelial cell, suggesting that the long term pharmacological inhibition of PAR-1 or its cleavage by blocking the activity of upstream proteases may have detrimental effects on PCa development [550]. In addition to thrombin-mediated cleavage of fibrinogen and PAR-1, it is important to note that thrombin cleaves various other substrates that influence the anti-tumor immune function such as FXIII, IL-1α, osteopontin, and the complementary factors. Although the development of PAR-targeting compounds has been challenging due to their wide distribution and diverse signaling, molecules inhibiting PAR-1, PAR-2, and PAR-4 cleavage and downstream signaling have been developed [551,552]. Particularly, PAR-2 has emerged as a valuable target to tackle AT thromboinflammation. Indeed, PAR-2 signaling drives DIO, FOXO-1-dependendent M1 macrophage polarization, and AT inflammation [61,553,554,555]. Importantly, human and rat obesity correlated with increased PAR-2 expression in the cells of the SVF, including macrophages, and PAR-2 antagonism was shown to reverse AT dysfunction and macrophage M1 polarization [61,553,554]. Nevertheless, PAR-2-deficient mice exhibit a lipodystrophy-like phenotype which has been, at the molecular level, characterized by impaired adipocyte differentiation, mitochondrial calcium signaling and mitochondrial biogenesis [556]. Therefore, PAR-2 antagonism in vivo may well impair adipocyte differentiation and AT development at basal conditions and thus, further investigation into the utility of PAR-2 as a therapeutic target is warranted.

### 10.2. UCP1-Dependent and UCP1-Independent Thermogenic Pathways

AT thermogenesis represents an extensively studied field, and harnessing the thermogenic potential of AT is perceived as a therapeutic strategy to curb the development of obesity and metabolic diseases [100,557,558]. Although this perspective stands true for relatively abundant visceral adipose pools, recent studies highlight the deleterious consequences of UCP1 upregulation in select adipose depots including the perivascular and the perirenal AT [559,560,561]. Therefore, modulating UCP1 expression and function has been proposed as a therapeutic strategy to counteract elevated hypoxia-associated AT inflammation [110,111]. In light of the increased interest in PPAT thermogenesis and its impact on the malignant behavior of PCa, investigating PPAT mitochondrial bioenergetics becomes pivotal for the therapeutic utility of the modulation of these pathways [109]. Indeed, as these thermogenic pathways often coexist but also present remarkable independency and spatial distinction, these pathways possibly respond differentially to thermogenic stimulation and present compensatory and alternative mechanisms [100,105,562,563,564]. Therefore, it becomes plausible that selectively targeting a particular thermogenic pathway in PPAT might mitigate AT bioenergetic dysfunction and its anticipated deleterious effect on PCa.

## 11. Conclusions

Several new aspects of therapeutic interventions emerged with the growing interest in the role of AT inflammatory changes in the pathogenesis of metabolic diseases and their complications. Thromboinflammation exists at the intersection between AT inflammation and the hypercoagulable state observed following metabolic deterioration in metabolic syndrome, obesity, and diabetes. The intersection of such cascades in the signaling networks contributing to PCa emergence and aggressiveness strongly suggests that the modulation of targets such as FXa, thrombin, and PARs might offer novel methods for halting the negative changes that underlie worsening of PCa associated with metabolic impairment. Indeed, there exists a dire need for meta-analysis of clinical trials encompassing different therapeutic approaches as well as anticoagulant therapies used in PCa patients in order to discover recommendations to guide clinicians’ decisions. Furthermore, extensive research into this emerging filed is required to examine the underlying molecular interactions in detail and refine the pharmacology of potential agents of value, including DOACs and possibly PAR ligands.

## Figures and Tables

**Figure 1 cancers-14-01679-f001:**
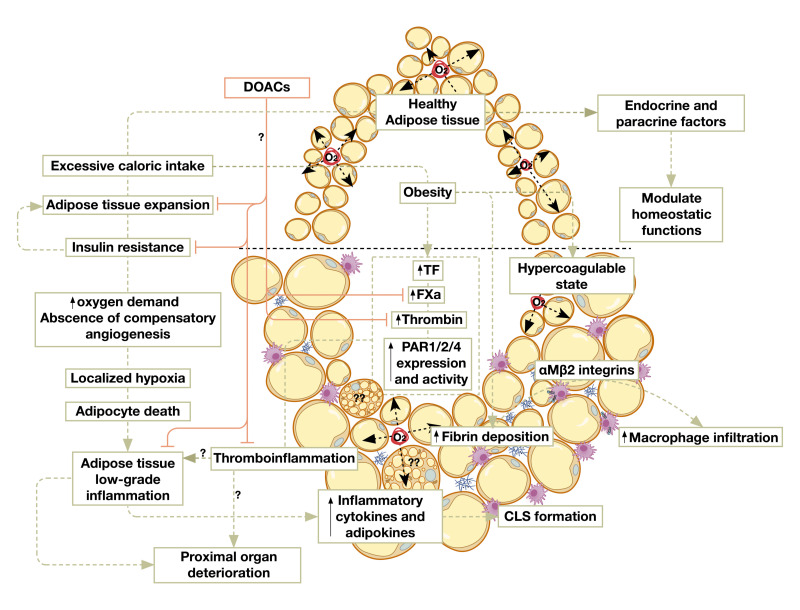
Activation of thromboinflammatory pathways following adipose tissue remodeling in metabolic syndrome and obesity. Excessive calorie intake and insulin resistance lead to adipocyte hypertrophy, thereby reducing adipose tissue oxygenation. The localized hypoxia evokes adipocyte death followed by macrophage infiltration. Beside the upregulation of typical inflammatory mediators, the same processes that trigger inflamed adipose milieu trigger a hypercoagulable state characterized by increased expression and activation of factor X (FXa) and thrombin, as well as different isoforms of the protease-activated receptors (PARs). These activated signaling cascades that are typically involved in blood clotting may lead to a further exacerbation of adipose tissue inflammation. Such mediators could act in a paracrine manner to cause proximal organ deterioration underlying the complications of metabolic disease. Interruption of the FXa or thrombin activity using direct oral anticoagulant (DOACs) drugs might therefore be useful in mitigating the adipose inflammatory state and ameliorating proximal organ damage.

**Figure 2 cancers-14-01679-f002:**
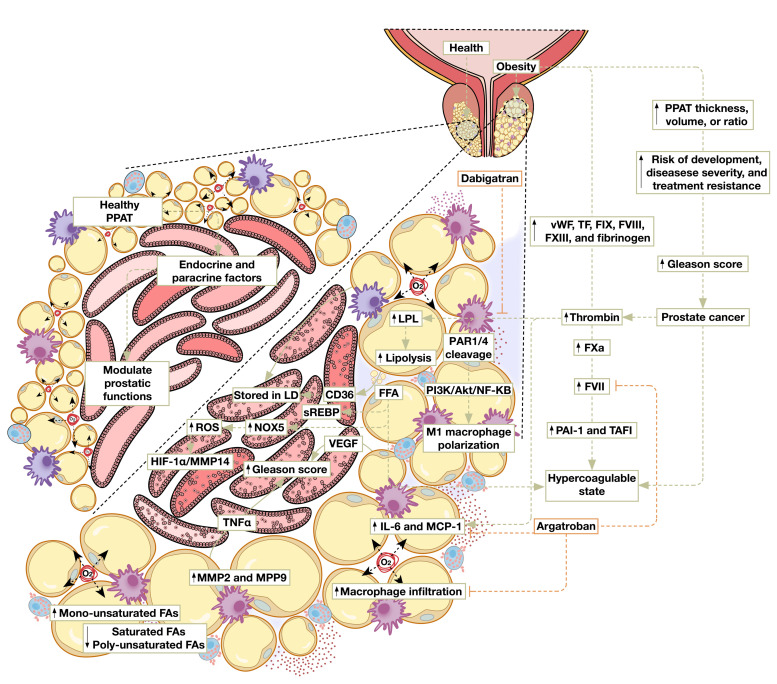
Periprostatic adipose tissue (PPAT) inflammation in obesity and metabolic syndrome worsens the prostatic cancer phenotype via several pathways involving activation of the thrombotic cascade. Increased PPAT thickness is associated with multiple changes, including the activation of the clotting pathway. In addition to prostatic cancer being a hypercoagulable state, further activation of thrombin evoked by the adipose inflammatory condition allows for increased macrophage infiltration and exacerbation of their shift to the M1 polarization with an increased production of proinflammatory cytokines. Moreover, thrombin evokes an increase in PPAT lipolysis with the resultant free fatty acids contributing to increased prostatic oxidative stress and inflammatory signaling. Evidence support a role for clotting factor inhibition in attenuating such changes.

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
