# Peer review of "Thromboinflammatory Processes at the Nexus of Metabolic Dysfunction and Prostate Cancer: The Emerging Role of Periprostatic Adipose Tissue"

_cancers, 2022, doi:10.3390/cancers14071679_

Round 1

Reviewer 1 Report

This review entitled "Thromboinflammatory Processes at the Nexus of Metabolic Dysfunction and Prostate Cancer: The Emerging Role of Periprostatic Adipose Tissue" is very important because the global prevalence of obesity, metabolic syndrome, and other metabolic disorders has increased today. Also, these metabolic disorders have been shown to increase the incidence of malignancies such as prostate cancer. Obesity, as a low degree of inflammation or chronic inflammation of adipose tissue, causes AT dysfunction and thromboinflammatory processes that occur in periprostatic adipose tissue and contribute to the development of prostate cancer and its progression. Good knowledge of the pathophysiological mechanisms that link these disorders may contribute to new therapeutic strategies for the treatment of prostate cancer.

The authors have used the latest literature in these areas, so this review paper is of great importance for understanding the mechanisms of metabolic dysfunction and prostate cancer, or thromboinflammatory processes that occur. To better understand these disorders, the authors have clearly shown in Figures 1 and 2.

Author Response

We would like to thank the reviewer for the supportive comments and for the interest in our work

Reviewer 2 Report

The manuscript by Alzhaim er al described the pivotal findings at the basis of the crosstalk between metabolic diseases such as diabetes and obesity and prostate cancer.

The review is well written and very interesting. Literature data appear well shown and the conclusion rather supported by data.

Nevertheless, other pieces of information would have probably further strengthened the conclusions of the manuscript:

  1. Please, provide a method section describing how the literature search was conducted and possibly summarize the search strategy in a flow diagram.
  2. Literature data on prostate cancer are continuously updated, so it could be informative to add a table describing therapeutic applications in clinical trials and a table describing clinically approved therapeutics.

3.The suggestion for the need of meta-analysis of clinical trials results as soon as possible in order to find recommendation to guide clinicians’ decisions seems to be relevant in the conclusion section.

  1. title must be more clear and concise
  2. English language must be improved
  3. please check the manuscript for grammar errors

Author Response

We thank the reviewer for the supportive and valuable comments:

1. Please, provide a method section describing how the literature search was conducted and possibly summarize the search strategy in a flow diagram.

Thank you very much for the comment. This was added as Section 2, Search Strategy.

2. Literature data on prostate cancer are continuously updated, so it could be informative to add a table describing therapeutic applications in clinical trials and a table describing clinically approved therapeutics.

We thank the reviewer for the comment. Since we mainly focus on the role of clotting factors as a potential link between periprostatic adipose inflammation and increased prostatic cancer aggressiveness, we feel that going into the details of the clinical data of the present therapies (likely to be unrelated to the topic at hand) will be out of the scope of the present article.

3. The suggestion for the need of meta-analysis of clinical trials results as soon as possible in order to find recommendation to guide clinicians’ decisions seems to be relevant in the conclusion section.

Thank you very much for the comment. We adopted your recommendation and adjusted the Conclusions section accordingly.

4. English language must be improved

5. please check the manuscript for grammar errors

Thank you very much for the comment. We have gone through the article in detail and fixed typos and grammatical errors.

Round 2

Reviewer 2 Report

The authors addressed all the points.